# Whole-body connectome of a segmented annelid larva

**Csaba Verasztó**[1,2†], **Sanja Jasek**[1,3†], **Martin Gühmann**[4],
**Luis Alberto Bezares-Calderón**[1], **Elizabeth A Williams**[5], **Réza Shahidi**[6],
**Gáspár Jékely**[1,3,5]*

[1]Living Systems Institute, University of Exeter, Exeter, United Kingdom; [2]École Polytechnique Fédérale de Lausanne (EPFL), Lausanne, Switzerland; [3]Heidelberg University, Centre for Organismal Studies (COS), Heidelberg, Germany; [4]School of Biological Sciences, University of Bristol, Bristol, United Kingdom; [5]BioSciences, University of Exeter, Exeter, United Kingdom; [6]Electron Microscopy Core Facility (EMCF), University of Heidelberg, Heidelberg, Germany

## eLife Assessment

This **important** study is an advancement towards the understanding of animal nervous system organization and evolution by providing an **exceptional**, high-quality and detailed description of the entire connectome of the 3-day larva of the marine annelid *Platynereis dumerilii*. It provides a wealth of data on cell type diversity and the modules that interconnect them. Its strength is the massive amount of high-quality data, although this is also partly a weakness as it can make the work difficult to read and digest scientifically. This work lays the foundations for studies on cell type diversity, segmental vs. intersegmental connectivity, and mushroom bodies, but will certainly also be of use to scientists interested in other nervous systems parts, their functions, and evolution.

**\*For correspondence:**
gaspar.jekely@cos.uni-heidelberg.de

[†]These authors contributed equally to this work

**Abstract** Nervous systems coordinate effectors across the body during movements. We know little about the cellular-level structure of synaptic circuits for such body-wide control. Here, we describe the whole-body synaptic connectome of a segmented larva of the marine annelid *Platynereis dumerilii*. We reconstructed and annotated over 9000 neuronal and non-neuronal cells in a whole-body serial electron microscopy dataset. Differentiated cells were classified into 202 neuronal and 92 non-neuronal cell types. We analyse modularity, multisensory integration, left-right, and intersegmental connectivity and motor circuits for ciliated cells, glands, pigment cells, and muscles. We identify several segment-specific cell types, demonstrating the heteromery of the annelid larval trunk. At the same time, segmentally repeated cell types across the head, the trunk segments and the pygidium suggest the serial homology of all segmental body regions. We also report descending and ascending pathways, peptidergic circuits, and a multimodal mechanosensory girdle. Our work provides the basis for understanding whole-body coordination in an entire segmented animal.

## Introduction

Nervous systems coordinate behaviour, physiology, and development through synaptic and neuroendocrine signalling. Signalling occurs specifically between groups of cells, organised into multilayered networks with precise synaptic and neuromodulatory connectivity (*Bentley et al., 2016*). Mapping such synaptic and chemical networks in blocks of neural tissue is the central aim of cellular-level connectomics (*Deng et al., 2019*; *Helmstaedter, 2013*; *Morgan and Lichtman, 2013*; *Williams et al., 2017*). For synaptic networks, connectomics requires volume imaging by serial electron microscopy (serial

EM) (*Schlegel et al., 2017*). The comprehensive analysis of whole-body coordination of actions by synaptic circuits would benefit from the cellular-level mapping of entire nervous and effector systems. Whole-animal synaptic connectomes have so far only been described for the nematode *Caenorhabditis elegans* (*Cook et al., 2019*; *White et al., 1986*) and the tadpole larva of the ascidian *Ciona intestinalis* (*Ryan et al., 2016*). Circuits spanning the entire central nervous system (CNS) have also been reconstructed in the larval CNS of the fruit fly *Drosophila melanogaster* (*Carreira-Rosario et al., 2018*; *Miroschnikow et al., 2018*; *Ohyama et al., 2015*) and in the three-day-old larva of the annelid *Platynereis dumerilii* (*Bezares-Calderón et al., 2018*; *Randel et al., 2015*; *Verasztó et al., 2017*). Recently, whole-brain connectomics has become possible in the larval and adult fly brain (*Franconville et al., 2018*; *Winding et al., 2023*; *Zheng et al., 2018*).

Here, we report the complete synaptic connectome and the cell-type complement of a 3-day-old larva (early nectochaete stage) of the marine annelid *Platynereis dumerilii*. This larval stage has three trunk segments, adult and larval eyes, segmental ciliary bands, and a well-developed somatic musculature (*Jasek et al., 2022*). The larvae show several behaviours, including visual phototaxis (*Randel et al., 2014*), UV avoidance (*Verasztó et al., 2018*), a startle response (*Bezares-Calderón et al., 2018*), and coordinated ciliary activity (*Verasztó et al., 2017*). The early nectochaete larva represents a transient dispersing stage in the life cycle of *Platynereis*. During this stage, the larvae do not feed yet but rely on maternally provided yolk. Compared to the juvenile and adult stages, it is expected that a considerable number of cell types will be only developing or completely missing at this stage. Three-day-old larvae do not yet have sensory palps and other sensory appendages (cirri); they do not crawl or feed and lack visceral muscles and an enteric nervous system (*Brunet et al., 2016*; *Williams et al., 2015*).

In *Platynereis* larvae, it has been possible to integrate behaviour with synapse-level maps, transgenic labelling of individual neurons, activity imaging, and gene knockouts (*Bezares-Calderón et al., 2018*; *Jokura et al., 2023*; *Verasztó et al., 2018*; *Verasztó et al., 2017*). Cellular-resolution gene expression atlases have also been developed for different larval stages. These can increasingly be integrated with single-cell transcriptomic atlases and synaptic circuit maps (*Achim et al., 2015*; *Vergara et al., 2021*; *Vergara et al., 2017*; *Williams et al., 2017*). The registration of a gene expression atlas to a whole-body low-resolution EM volume in the 6-day-old *Platynereis* larva has also been reported (*Vergara et al., 2021*).

We previously reported synaptic connectomes for several whole-body circuits from a transmission EM volume of a 3-day-old larva (*Randel et al., 2015*). These include the visual, startle, ciliomotor, nuchal organ, and neurosecretory systems (*Bezares-Calderón et al., 2018*; *Randel et al., 2015*; *Shahidi et al., 2015*; *Verasztó et al., 2018*; *Verasztó et al., 2017*; *Williams et al., 2017*). Here, we report the complete synaptic connectome and the cell-type complement from the same volume of a 3-day-old *Platynereis* larva. Recently, we also reported the desmosomal connectome and motoneuron innervation of the somatic musculature (*Jasek et al., 2022*) and a preprint of a preliminary analysis of the synaptic connectome (*Verasztó et al., 2020*) that this work supersedes. The full connectome reconstruction has now allowed us to uncover several new circuits and to consider all circuits in a whole-body context. We also found several neurons that span the entire length of the larva, highlighting the strength of a whole-body dataset. These cells and their circuits allow us to generate hypotheses on how whole-body coordination may be achieved by the larval nervous system. The connectome also allows us to address long-standing hypotheses about the origin of the segmented annelid body plan and explore patterns of circuit evolution.

## Results
### Serial EM reconstruction of a *Platynereis* larva

We traced and annotated all cells in a previously reported volume EM dataset of a 3-day-old (72 hours post fertilisation (hpf)) *Platynereis* larva (*Randel et al., 2015*; *Figure 1A*, *Figure 1—figure supplement 1*). The dataset consists of 4846 layers of 40 nm thin sections scanned by transmission electron microscopy (TEM). The sections span the entire body of the larva. In the volume, we identified 9162 cells with a soma.

We skeletonised cells containing projections or having an elongated morphology, including muscle cells, glia, and neurons (*Figure 1C and D*). During skeletonisation, interconnected nodes are placed

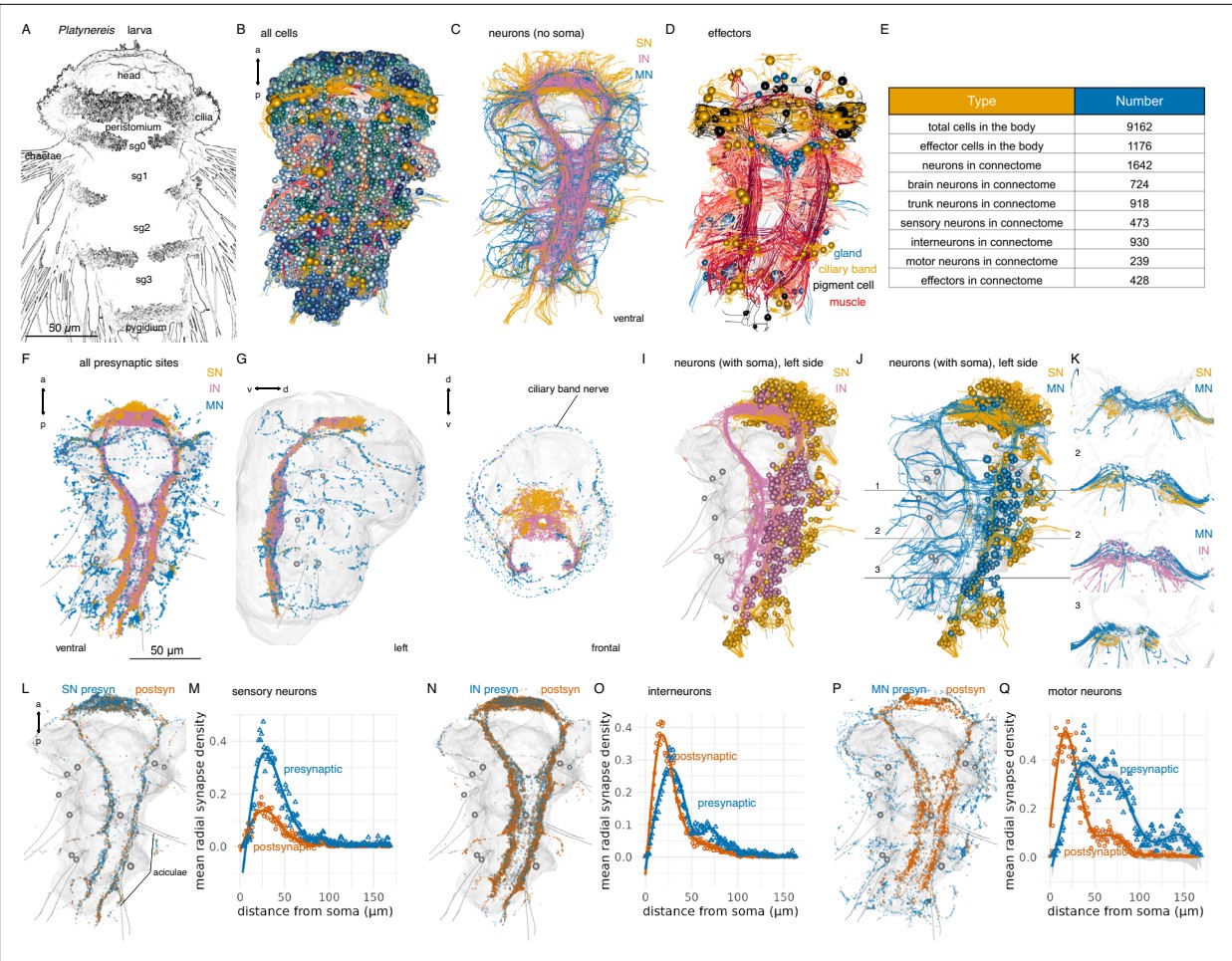

**Figure 1.** Overview of nervous-system anatomy in the 3-day-old *Platynereis* larva. (**A**) Stylised scanning electron microscopy image of a 3-day-old segmented *Platynereis* larva. (**B**) Morphological rendering of all cells in the electron microscopy (EM) volume. Spheres represent the position of cell somas. (**C**) Neurite processes of all neurons in the larva coloured by neuron type (ventral view). Soma are not shown. (**D**) All effector cells in the larva, including ciliated cells (yellow), glands (grey), and muscles (red). Spheres indicate cell soma (not shown for muscle cells) (**E**) Summary of cell numbers of different categories in the larva. (**F**–**H**) All presynaptic sites coloured by neuron type shown in (**F**) ventral, lateral (**G**), and frontal (**H**) views. (**I**) Morphological rendering of all sensory and interneurons on the left side of the larva. (**J**) Morphological rendering of all sensory and motor neurons on the left side of the larva. Horizontal lines indicate the position of the cross-sections in (**K**). (**K**) Cross-section view of sensory, inter-, and motor neuron projections in the ventral nerve cord neuropil. Cross-sections at three antero-posterior positions are shown. Numbers indicate the position of the cross-sections as marked in (**J**). (**L**) Position of presynaptic and postsynaptic sites on all sensory neurons of the connectome. (**M**) Mean radial synapse density (radius: 1000 nm, centre: soma) of presynaptic and postsynaptic synapses in sensory neurons. (**N**) Position of presynaptic and postsynaptic sites on all interneurons of the connectome. (**O**) Mean radial synapse density (radius: 1000 nm, centre: soma) of presynaptic and postsynaptic synapses in interneurons. (**P**) Position of presynaptic and postsynaptic sites on all motor neurons of the connectome. (**Q**) Mean radial synapse density (radius: 1000 nm, centre: soma) of presynaptic and postsynaptic synapses in motor neurons. In all anatomical renderings, the outline of the yolk is shown in grey. In (**G**), the body outline is also shown. Aciculae and chaetae are also shown in grey as segmental markers. Abbreviations: SN, sensory neuron; IN, interneuron; MN, motor neuron, sg0-3, segments 0–3. *Figure 1—source data 1*. Source data for panels **M**, **O**, **Q**.

The online version of this article includes the following source data and figure supplement(s) for figure 1:

**Source data 1.** Mean radial synapse density data.

**Figure supplement 1.** Anatomy of the 3-day-old *Platynereis dumerilii* larva.

**Figure supplement 2.** Morphological parameters of neurons.

**Figure supplement 2—source data 1.** Source data for all plots in *Figure 1—figure supplement 2*.

in the neurite cross-section profiles of the same neuron across layers. The skeletons are grown until all branches of a neuron have been traced. The individual nodes can be tagged and the skeletons named and multiply annotated. For all neuronal skeletons, we also identified and marked synaptic sites as connectors. Connectors link a node of a presynaptic skeleton to a node of a postsynaptic skeleton partner. Synapses were recognised as presynaptic vesicle clusters at the membrane with no postsynaptic specialisations, as previously described (*Randel et al., 2014*). Synapses are monadic, connecting one presynaptic neuron to one postsynaptic cell. Querying all skeletons and synaptic connectors allowed us to derive the synaptic connectome (*Figures 1 and 2*). We did not identify any gap junctions.

The skeletons across the volume comprised 5,661,050 nodes and had 28,717 presynaptic and 27,538 postsynaptic sites. We could not attach 15,020 fragments (896,428 nodes) to a skeleton with a soma. These fragments contained 4122 presynaptic and 5304 postsynaptic sites. Most of the fragments represent short skeletons of twigs (median length 1.04 μm) (*Figure 1—figure supplement 2B*) that could not be traced across gaps or low-quality layers. Overall, 15.8% of all nodes, 14.4% of presynaptic, and 19.3% of postsynaptic sites are on fragments and not assigned to a cell with a soma.

## Neuroanatomy and neuropils

The *Platynereis* larval nervous system is subdivided into an anterior brain and a rope-ladder-like paired ventral nerve cord (VNC). The brain and VNC are connected by circumesophageal connectives (*Figure 1C and F–H*). The trunk has five segments, an anterior cryptic segment (segment 0) (*Saudemont et al., 2008*; *Steinmetz et al., 2011*), three main segments (segments 1–3) with parapodia, and a posterior pygidium (*Figure 1A and B*).

Morphological rendering of all synapses in the volume highlights the overall organisation of neuropils and motor nerves. In the brain, sensory, inter-, and motoneurons synapse in partly overlapping regions, with sensory neuron synapses occurring more anteriorly. In the head as well as the VNC, motor synapses mark the periphery (*Figure 1F–H*).

The VNC also has an organised neuropil architecture. Sensory neurons in the trunk only have ipsilateral projections that concentrate in two ventro-lateral bundles in the VNC (*Figure 1I–K*). Interneurons have either ipsi- or contralateral projections that remain in the VNC neuropil and also concentrate ventrally in the VNC (*Figure 1K*). Motoneurons can have ipsilateral and contralateral projections and are the only neuron class with peripheral afferent projections (*Figure 1J*). Motor nerves concentrate dorsally in the VNC (*Figure 1K*), close to the basal lamina and mesodermal muscles.

Neurons have a median cable length of 97.1 μm (sensory neurons: 101.6 μm; interneurons: 87.77 μm, motoneurons: 139.8 μm) (*Figure 1—figure supplement 2A*) and the largest neurons (Ser-tr1 and Loop) have a cable length of 1.2–1.7 mm.

Neurons have a median of eight presynaptic and six postsynaptic sites with the largest neurons (MC, Ser-tr1, Loop) with over 300 presynapses (*Figure 1—figure supplement 2*).

The relative ratio of post- and presynaptic sites distinguishes sensory, inter-, and motoneurons, with sensory neurons having relatively more presynapses (*Figure 1—figure supplement 2E*).

The number of pre- and postsynaptic sites and skeleton segments scales with the total cable length of skeletons with Pearson's correlation coefficients >0.7 (*Figure 1—figure supplement 2F–H*).

Most neuronal trees are unipolar and not highly branched (<100 segments; *Figure 1—figure supplement 2H*). Neurons in general do not have primary cilia. Sensory neurons are bipolar with an axon and a sensory dendrite bearing zero to five sensory cilia at its distal tip, showing different sensory specialisations (*Bezares-Calderón et al., 2018*; *Williams et al., 2017*). Interneurons are unipolar or pseudo-unipolar. Ciliomotor neurons have two motor axons emanating from the soma (*Verasztó et al., 2017*); other motoneurons are unipolar.

## Derivation and network analysis of the whole-body synaptic connectome

To define a whole-body synaptic connectome for the larva, we comprehensively identified synapses by traversing the volume and each skeleton multiple times (32,381 synapses). Synapses were connected to the pre- and postsynaptic skeletons (one to one, as synapses are monadic). We then retrieved all synapses and their pre- and postsynaptic skeletons and derived a graph (6725 graph nodes or vertices). From this graph, nodes with less than three connections were removed. The final synaptic

connectome contains 2675 nodes (including 467 fragments) connected by 14,066 directed edges formed by 26,881 in-graph synapses (*Figure 2*). The connectome is a sparsely connected network with a graph density of 0.00197.

In the graph, the majority of source nodes (nodes with only outgoing edges) are sensory neurons, sink nodes (nodes with only incoming edges) are mostly effector cells (*Figure 2—figure supplement 1*). The nodes with the highest degree included ciliomotor neurons (e.g. Loop, Ser-tr1) (*Bezares-Calderón et al., 2018*; *Verasztó et al., 2017*), the sensory-motoneuron pygPBunp (*Verasztó et al., 2017*), and the motoneurons of exocrine glands (MNspinning, see below). Some of the strongest edges (highest number of synapses) were between the pigment-cell motoneuron cioMNcover and prototroch pigment cells, MNspinning and exocrine glands, and the previously described MC cell and ciliated cells (*Verasztó et al., 2017*).

To identify more strongly connected subgraphs within the connectome, we used community detection with the Leiden algorithm (*Traag et al., 2019*). This analysis, combined with force-field-based clustering, highlighted several modules (*Figure 2*). Nodes in a module are more strongly interconnected than to nodes in other modules (*Figure 2—figure supplement 2*). We named the modules based on their primary effector organs or other dominant anatomical characters. We would like to note that this analysis can recover varying numbers of modules depending on the resolution parameter (*Figure 2—figure supplement 2C*). The selected resolution parameter subdivides the network into communities that we think best represent functional units.

The modules include the anterior neurosecretory centre, the mushroom bodies, the visual circuit, a network of ciliomotor neurons, a module innervating head pigment cells, a left and right mechanosensory module providing input to muscles and segmental exocrine glands, a module for trunk postural control, and five muscle-motor modules (*Figure 2*).

The modules contain neurons that project to distinct neuropil domains (*Figure 2—figure supplement 3*) and likely correspond to functional units. The visual module contains neurons of the eyes and primary visual neuropil (PRC, IN1, INint, INton) previously shown to mediate visual phototaxis (*Randel et al., 2014*). The ciliomotor module contains ciliary band cells and functionally related neurons that show coordinated activity and drive ciliary closures and beating (Ser-tr1, Loop, MNant) (*Verasztó et al., 2017*). The anterior neurosecretory centre includes anterior and dorsal sensory neurons, the ciliary photoreceptors and their postsynaptic circuit, including the INRGWa and INNOS interneurons, as well as the Ser-h1 and MC ciliomotors with their prototroch targets. Neurons in this module mediate barotaxis and UV avoidance (*Calderón et al., 2023*; *Jokura et al., 2023*; *Verasztó et al., 2018*).

The modules are also interconnected among themselves suggesting crosstalk. Most sensory neurons occur in the visual, the two mechanosensory, the mushroom body, and the anterior neurosecretory modules (*Figure 2—figure supplement 1J*). In a Sankey network diagram representing information flow, these modules are upstream of the motor-dominated pigmentmotor, ciliomotor, and musclemotor modules (*Figure 2—figure supplement 2B*).

## Cell-type classification and neuronal diversity

In the volume, each skeleton was multiply annotated. Neurons and non-neuronal cells were categorised into classes (sensory, inter- and motoneurons, muscle, gland, etc.) and further subdivided into cell types.

For classifying neurons into cell types, we used a combination of five morphological and connectivity criteria: (i) position of cell somata, (ii) the morphology of neurite projections (e.g. branching pattern, decussating, ascending, or descending), (iii) the ultrastructure of sensory specialisations (e.g. number and type of cilia, microvilli — for sensory neurons only), (iv) neuropeptide content as determined by the siGOLD immunolabelling method (*Shahidi et al., 2015*), and (v) synaptic connectivity. Most cell types show left-right symmetry except for a few asymmetric or midline neurons (e.g. SN_YF5cil, pygPBunp).

Based on these criteria, we classified 966 neurons into 202 cell types (*Supplementary file 1*, *Video 1*, *Figure 3*). Most neuronal cell types are represented by only two cells in the entire body (*Figure 3D*).

We also categorised the remaining 8196 cells in the larva. Of these, 3128 were classified into 92 non-neuronal cell types (*Supplementary file 1*, *Figure 3—figure supplement 1*).

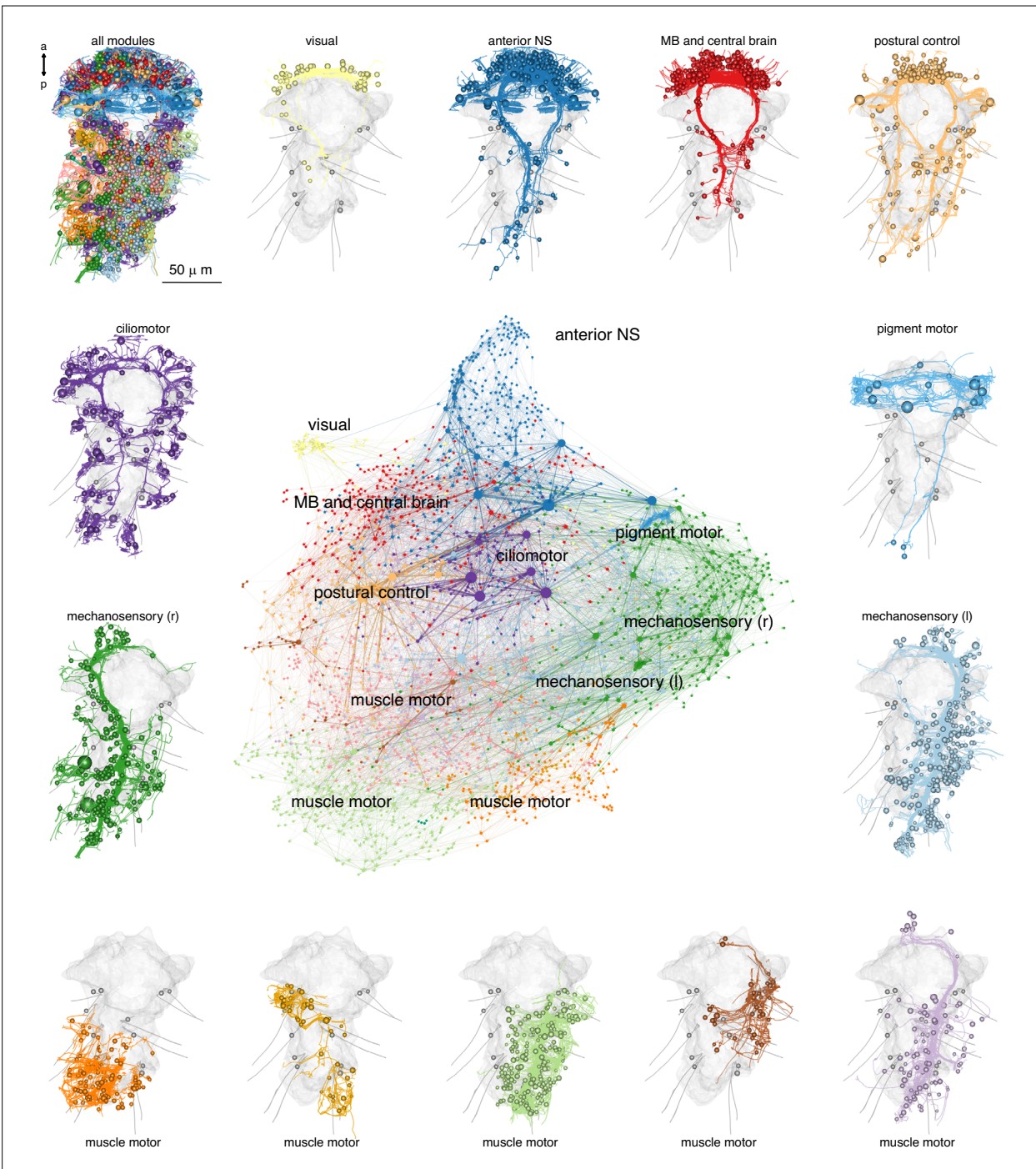

**Figure 2.** Modularity of the *Platynereis* larval connectome. Full connectome graph of the *P. dumerilii* 3-day-old larva chemical synapse connectome. Nodes represent individual cells, edges represent synaptic connectivity. Nodes are coloured by modules. Node sizes are proportional to weighted degree. The individual panels show the morphology of all cells in each module with the outline of the yolk and aciculae shown in grey. *Figure 2—source data 1*. Source data of the connectome graph in tibble graph (tbl_graph) format saved as an R binary object. For an interactive version of the graph, see also https://jekelylab.github.io/Platynereis_connectome/Full_connectome_modules_with_names.html (a permanent version is in *Verasztó et al., 2025* in /supplements/celltype_compendium_website).

The online version of this article includes the following source data and figure supplement(s) for figure 2:

**Source data 1.** Source data of the connectome graph in tibble graph (tbl_graph) format saved as an R binary object.

**Figure supplement 1.** Network parameters of the connectome.

**Figure supplement 2.** Connectivity of connectome modules.

*Figure 2 continued on next page*

*Figure 2 continued*

**Figure supplement 2—source data 1.** Source data of the connectivity matrix of the connectome modules saved as a csv file.

**Figure supplement 3.** Neuropils formed by connectome modules.

These included epidermal cells (1334 cells), various pigment cells (158 cells of seven types), muscle cells (853 cells of 53 types, described in *Jasek et al., 2022*), locomotor ciliated cells (80 cells of six types, described in *Verasztó et al., 2017*), glial cells (78 cells of three types), gland cells (54 cells of six types), various support and sheet cells, nephridia, putative migratory cells, parapodial cells producing chitin chaetae (106 cells) or aciculae (12 cells), and the follicle cells (566 cells of five types) ensheathing them (*Supplementary file 1*; *Video 1*; *Figure 3—figure supplement 1*).

In addition, we defined 18 broader neuronal cell groups, containing neurons of similar morphology (e.g. head decussating neurons) but in either differentiated or immature state (e.g. immature palp sensory neurons).

The annotations are hierarchical and cell groups can contain one or more differentiated cell types (*Supplementary file 1*). The remaining 5068 cells are either dividing cells (68 cells), undifferentiated cells that putatively belong to the neuronal lineage (1597 cells) and various developing neurons with projections but no or only very few synapses (2457). These cells were not classified into cell types.

The developing antennae contain 115 cells of which the majority have only a few or no synapses and immature sensory dendrites. In the developing palps, we found 82 developing sensory neurons. The developing mouth or stomodeum (675 cells) is lined with 52 immature sensory neurons.

All cells were annotated in CATMAID (*Saalfeld et al., 2009*; *Schneider-Mizell et al., 2016*) with information representing the above categories. We also annotated all cells based on their soma position in the body (e.g. left or right side, segment 0–3, germ layer etc.) (*Figure 3—figure supplement 2*). These annotations were used to query the data and visualise subsets of cells.

*Supplementary file 1* lists all neuronal and non-neuronal cell types, their main annotations, including morphological, cell-class, segment, ganglion or sensory organ, neurotransmitter and neuropeptide phenotype, and main synaptic partners. We also listed all literature references for previously published neuronal and non-neuronal cells and indicated which cells are reported here for the first time (105 cell types out of 294). Interactive 3D morphological renderings of each cell type together with their main annotations can also be explored on a webpage (https://jekelylab.github.io/Platynereis_connectome/; permanent version in *Verasztó et al., 2025*, /supplements/celltype_compendium_website). Querying by two or more annotations allows tallying the number of cells in different category (e.g. number of sensory cells with a single penetrating cilium or the number of gland cells in a particular segment) (*Figure 3—figure supplement 3*).

Left-right cell-type pairs have similar arbour morphologies (*Figure 3—figure supplement 4*). The similar branching patterns for left and right cells of the same type are also indicated by their similar average Sholl diagrams (*Figure 3—figure supplement 5*). Left-right pairs also have projections at symmetrical positions in the VNC indicating neuropil stereotypy (*Figure 3—figure supplement 6*).

Presynaptic and postsynaptic sites in a neuron can be intermingled or spatially segregated. Interneurons can have mixed or spatially segregated input-output compartments. Most motoneurons have spatially segregated input-output compartments (*Figure 1L-Q*, *Figure 3—figure supplements 7 and 8*).

## Grouped cell-type-level connectome

The cell-type classification allowed us to analyse a grouped synaptic connectivity graph where cells of the same type were collapsed into one node and synapse counts were summed (*Figure 4A*).

This cell-type connectome included 84 sensory, 82 interneurons, 34 motoneuron types, and 58

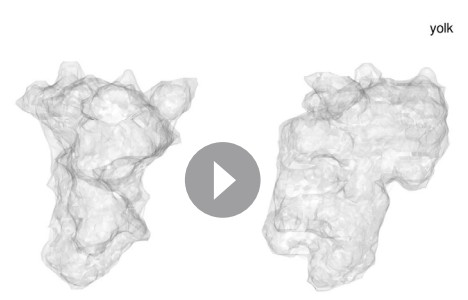

yolk

**Video 1.** Morphological rendering of all cells and synapses in the 3-day-old larva, organised by class, ventral (left side) and left-later (right side) views.
https://elifesciences.org/articles/97964/figures#video1

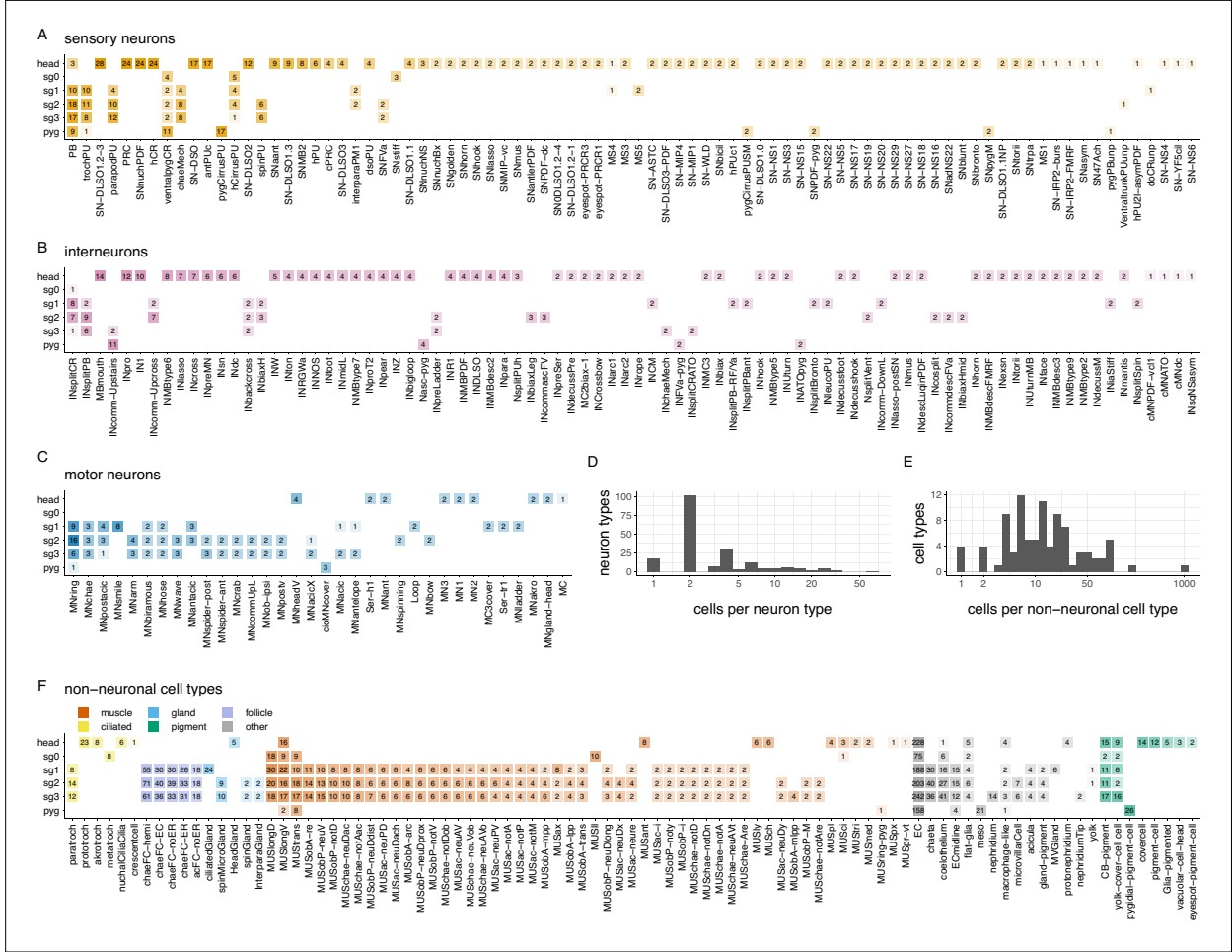

**Figure 3.** Cell-type complement of the 3-day-old *Platynereis* larva. Segmental distribution and number of sensory neuron types (**A**), interneuron types (**B**), and motoneuron types (**C**). Histogram of the number of cells per neuronal (**D**) and non-neuronal (**E**) cell types. (**F**) Segmental distribution and number of non-neuronal cell types. The number in each box refers to the number of cells of the indicated cell type in the indicated segment. *Figure 3—source data 1*. Source data for panels **A**–**C**. *Figure 3—source data 2*. Source data for panel **F**.

The online version of this article includes the following source data and figure supplement(s) for figure 3:

**Source data 1.** Source data for panels A–C containing the cell-type classification and number per segment of sensory, inter and motor neurons.

**Source data 2.** Source data for panel F containing the classification and number per segment of non-neuronal cell types.

**Figure supplement 1.** Major cell classes in the 3-day-old *Platynereis* larva.

**Figure supplement 2.** Major annotations of cell types.

**Figure supplement 2—source data 1.** The annotation matrix in txt format with cell types ordered by their annotation (celltype1-celltype202).

**Figure supplement 3.** Number of skeletons for double annotations.

**Figure supplement 3—source data 1.** The number of skeletons for pairs of annotations, data in tidy format.

**Figure supplement 4.** Example neuronal cell types.

**Figure supplement 5.** Average Sholl diagrams for all symmetrical neuronal cell types.

**Figure supplement 5—source data 1.** Average Sholl values for all symmetrical neuronal cell types.

**Figure supplement 6.** Position of axons of left-right symmetric neuron pairs.

**Figure supplement 7.** Radial density of presynaptic and postsynaptic sites in head neurons.

**Figure supplement 7—source data 1.** Radial density data of incoming (postsynaptic) and outgoing (presynaptic) synapses for all 202 cell types.

**Figure supplement 8.** Radial density of presynaptic and postsynaptic sites in trunk neurons.

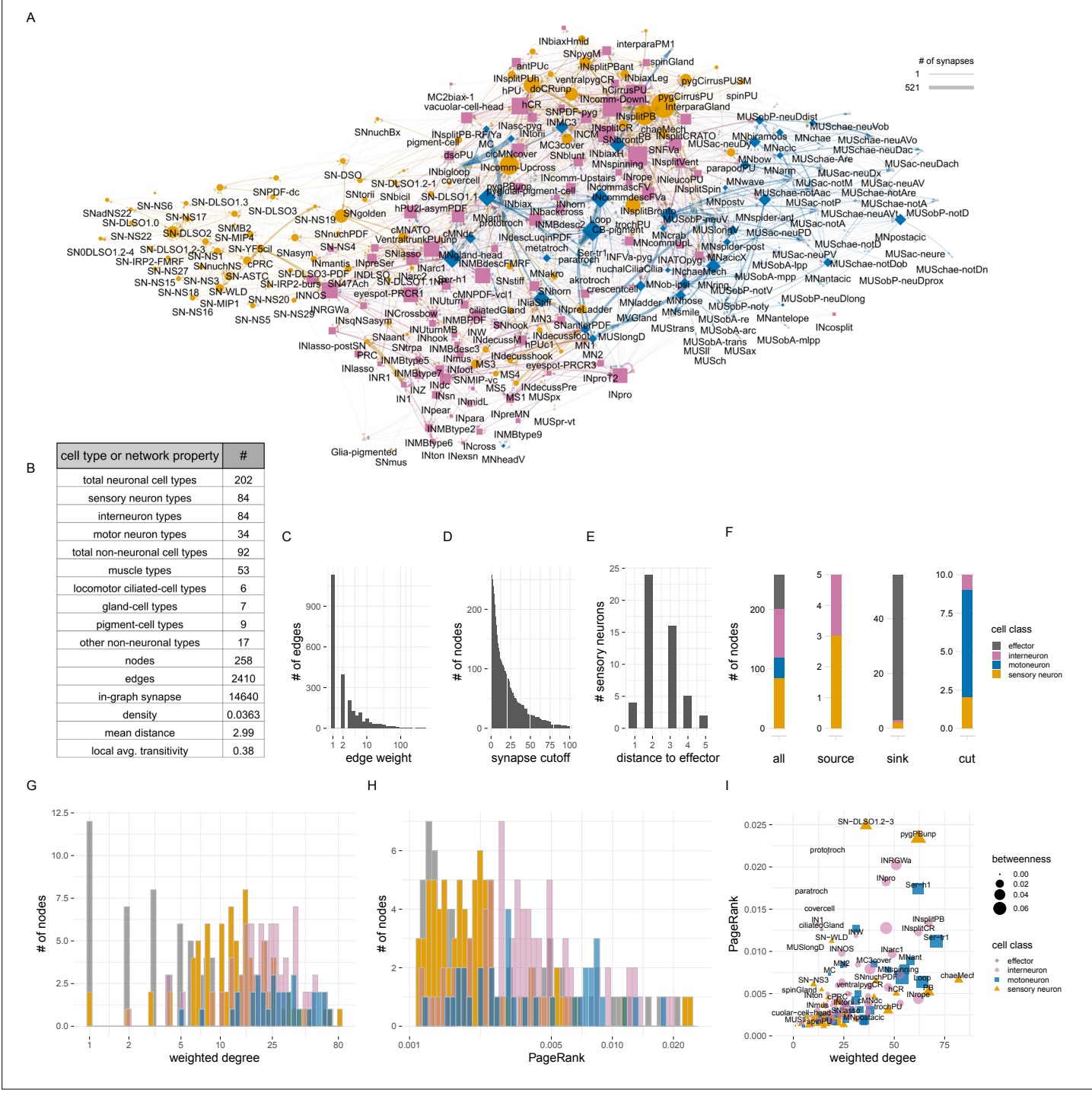

**Figure 4.** Cell-type-level connectivity of the *Platynereis* larval connectome. (**A**) The cell-type connectome of the 3-day-old larva. Nodes represent grouped cells of the same type, edges represent synaptic connectivity (square root of the sum of synapses). Nodes are coloured by cell class (SN - orange; IN - magenta; MN - blue; effector - grey). (**B**) Table of cell type and network statistics. (**C**) Histogram of edge weights. (**D**) Size of the largest network after removing edges of increasing weights. (**E**) Number of sensory neurons with different path distances from effector cells. (**F**) Number of SN, IN, MN, and effector cell types in the cell-type-level connectome shown for all nodes, source nodes, sink nodes, and cut nodes. (**G**) Histogram of weighted degree of nodes, plotted for each cell class. (**H**) Histogram of pagerank of nodes, plotted for each cell class. (**I**) Weighted degree of nodes in the cell-type connectome in relation to node pagerank-centrality. *Figure 4—source data 1*. Source data of the cell type connectivity graph in tbl_graph format saved as an R binary file. For an interactive version of the graph, see also https://jekelylab.github.io/Platynereis_connectome/Figure4_celltype_network.html.

*Figure 4 continued on next page*

*Figure 4 continued*

The online version of this article includes the following source data and figure supplement(s) for figure 4:

**Source data 1.** Source data of the cell type connectivity graph in tbl_graph format saved as an R binary file.

**Figure supplement 1.** The *Platynereis* larval connectome grouped by cell type.

**Figure supplement 2.** Comparison of connectivity between the left and right body sides.

**Figure supplement 2—source data 1.** Left-right cell-type connectivity in tibble format.

**Figure supplement 3.** Comparison of connectivity between the left and right body sides.

**Figure supplement 3—source data 1.** Source data for the Pearson correlation data for the left-right cell-type comparisons.

**Figure supplement 4.** Network properties of the *Platynereis*, C. *elegans,* and Ciona connectomes in comparison to a random graph.

**Figure supplement 5.** Grouped connectivity graph.

**Figure supplement 5—source data 1.** The celltype network in visNetwork format saved as an R RDS source file, saved with zip compression.

effector cell types (including ciliary band, muscle, gland, and pigmented cell types) (*Figure 4—figure supplement 1*).

The cell-type graph had 2410 edges formed by 14,640 synapses (*Figure 4*).

The correlation of the left and right synapse matrices was 0.91, indicating a high level of stereotypy of connectivity (*Figure 4—figure supplement 2*, *Figure 4—figure supplement 3*).

The cell-type graph is characterised by network parameters similar to the *C. elegans* and *C. intestinalis* connectome graphs (*Figure 4—figure supplement 4*), suggesting a similar overall circuit organisation at the level of cell types, but with an order of magnitude more cells in *Platynereis*.

To explore sensory-motor pathways in the grouped graph, we searched for the shortest directed paths from all sensory neurons to all effector cells. We identified four sensory neurons with direct motor output (sensory-motor neurons: eyespot-PRCR3, pygPBunp, pygCirrusPUSM, hPU2l-asymPDF) and 25 premotor sensory neurons. The maximum number of hops from sensory neurons to an effector is five. 33 sensory neurons have no synaptic paths to effectors. 19 of these are neurosecretory cells with few or no synaptic partners (*Williams et al., 2017*; *Table 1*). For a graph-based visualisation of information flow, we arranged the nodes according to their relative ratio of post- and presynapses (*Figure 4—figure supplement 5*). This layout indicates that sensory-evoked activity propagates from sensory neurons through interneurons and motoneurons to effectors.

## Neurotransmitter and neuropeptide phenotypes

We could assign neurotransmitters or neuropeptides to 53 (26%) neuronal cell types (*Figure 5*). This was possible either by direct immunogold labelling for neuropeptides (*Shahidi et al., 2015*) or by matching the position and morphology of cells to whole-body gene expression data or neuronal transgenic reporter expression (*Randel et al., 2014*; *Verasztó et al., 2017*; *Vergara et al., 2017*).

We annotated 11 cholinergic, three serotonergic, one dopaminergic, one adrenergic, and four glutamatergic neuronal cell types. In addition, we assigned one of 12 neuropeptides to 38 cell types (pigment dispersing factor — 12, allatotropin/orexin — 5, leucokinin — 1, proenkephalin — 1, FVamide — 5, FMRFamide — 3, myoinhibitory peptide — 3, achatin — 1, RGWamide — 1, MLD/pedal peptide — 1, IRP2 — 1, WLD — 1). Neuropeptides occur in sensory, motor, and interneurons.

**Table 1.** Sensory neurons categorised by path length to effectors.

| Sensory-motor | Premotor SN | SN 2-5 hops from effectors | SN with no path to effectors |
|---|---|---|---|
| eyespot-PRCR3, pygPBunp, pygCirrusPUSM, hPU2l-asymPDF | SNnuchPDF, SNhorn, SNhook, SNlasso, SNMIP-vc, SNantlerPDF, eyespot-PRCR3, MS1, MS4, MS5, pygPBunp, chaeMech, PB, antPUc, spinPU, trochPU, parapodPU, hPU, dsoPU, hCR, ventralpygCR, SNFVa, SNblunt, SNbronto, SNtorii | PRC, cPRC, SNnuchPDF, SNnuchNS, SNnuchBx, SNhorn, SNhook, SNlasso, SNMIP-vc, SNantlerPDF, eyespot-PRCR3, eyespot-PRCR1, MS1, MS4, MS5, SN-ASTC, SN-IRP2-burs, SN-DSO, SNbicil, SN47Ach, pygPBunp, chaeMech, PB, interparaPM1, antPUc, spinPU, trochPU, pygCirrusPUSM, hCirrusPU, parapodPU, hPU, dsoPU, pygCirrusPU, hPU2l-asymPDF, hCR, ventralpygCR, doCRunp, SN-NS15, SNaant, SNPDF-pyg, SN-NS5, SN-YF5cil, SN-NS16, SNFVa, SNblunt, SN-DLSO1.3, SNstiff, SNbronto, SNpygM, SNtorii, SNtrpa | SNgolden, SNmus, SNPDF-dc, SN0DLSO1.2–4, SN-DLSO1.2–3, SN-DLSO1.2–1, SN-DLSO2, SNMB2, MS3, SN-DLSO3-PDF, SN-DLSO3, SN-IRP2-FMRF, SN-MIP4, SN-MIP1, SN-WLD, SNasym, hPUc1, SN-DLSO1.0, SN-NS1, SN-NS3, SN-NS4, SN-NS22, SN-NS17, SN-NS19, SN-NS20, SN-NS6, SN-NS29, SN-NS27, SN-NS18, SNadNS22, SN-DLSO1.1, SN-DLSO1.1NP, VentraltrunkPUunp |

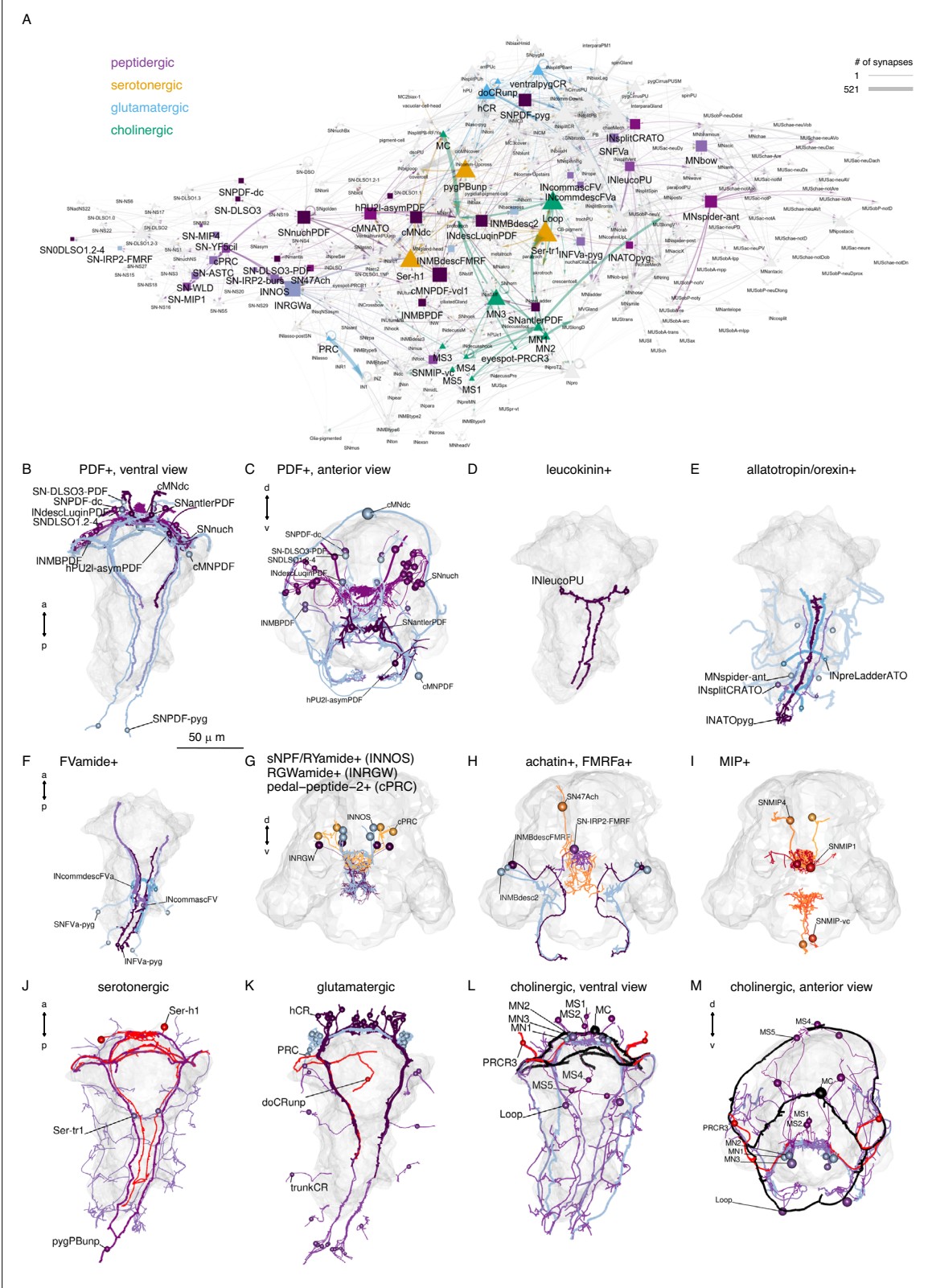

**Figure 5.** Transmitter phenotypes mapped at single-cell resolution. (**A**) The cell-type connectome graph with nodes coloured based on neurotransmitter phenotype. (**B**, **C**) Morphological rendering of neurons with immunogold labelling for pigment dispersing factor (PDF) neuropeptide, ventral (**B**) and anterior (**C**) views. (**D**) Neurons with immunogold labelling for leucokinin. (**E**) Neurons with immunogold labelling for allatotropin/orexin. (**F**) Neurons with immunogold labelling for FVamide. (**G**) Neurons with immunohistochemically mapped expression of sNPF/RYamide, RGWamide, and pedal

*Figure 5 continued on next page*

*Figure 5 continued*

peptide 2/MLD neuropeptides. (**H**) Neurons with immunohistochemically mapped expression of achatin and immunogold- or immunohistochemically-mapped (SN-IRP2-FMRF) FMRFamide neuropeptide. (**I**) Neurons with immunogold (SNMIP-vc) or immunohistochemically mapped expression of myoinhibitory peptide (MIP) expression. (**J**) Neurons with immunohistochemically mapped expression of serotonin and genetically mapped expression of tryptophan hydroxylase (TrpH), a serotonergic marker. (**K**) Neurons with genetically mapped expression of vesicular glutamate transporter (VGluT), a glutamatergic marker. (**L, M**) Neurons with genetically mapped expression of vesicular acetylcholine transporter (VAChT) and choline acetyltransferase (ChAT), cholinergic markers, ventral (**L**) and anterior (**M**) views. *Figure 5—source data 1*. Interactive HTML file with the network shown in panel **A**. For an interactive version of the graph, see also https://jekelylab.github.io/Platynereis_connectome/network_with_transmitters.html.

The online version of this article includes the following source data and figure supplement(s) for figure 5:

**Source data 1.** Interactive HTML file with the network shown in panel A, saved with zip compression.

**Figure supplement 1.** Neurons with dense-cored vesicles.

**Figure supplement 2.** Circuits of PDF, allatotropin/orexin, and leucokinin neurons.

**Figure supplement 2—source data 1.** Source data of the network diagram of PDF-expressing neurons and their pre- and postsynaptic partners.

**Figure supplement 2—source data 2.** Source data of the network diagram of allatotropin/orexin- and leucokinin-expressing neurons and their pre- and postsynaptic partners.

In agreement with the large number of neuropeptides expressed in the larva (*Conzelmann et al., 2011*; *Shahidi et al., 2015*; *Williams et al., 2017*), 516 connectome neurons contained dense core vesicles, indicative of peptidergic transmission (*Figure 5—figure supplement 1*).

We highlight cellular and sensory-effector circuit examples for PDF, leucokinin, and allatotropin/orexin neuropeptides (*Figure 5—figure supplement 2*). For example, two leucokinin-expressing interneurons (INleucoPU) in the first trunk segment are postsynaptic to mechanosensory PU cells and INsplitPB neurons and synapse on MNladder, MNcommUpL, and MNwave motoneurons.

These neuropeptide-expressing cells and their mini-circuits pinpoint potential sites of peptidergic modulation mapped to single-cell resolution within the whole-body connectome.

## Brain ganglia, neuropils, cell types and circuits

The annelid nervous system has a ganglionic architecture. In the head of the *Platynereis* larva, the different head sensory neurons form several ganglionic clusters, including a median-dorsal sensory cluster, an apical organ, a dorso-lateral sensory cluster, the adult eyes and nuchal organs, the antennae, the palps, and the ventro-lateral mushroom bodies (*Video 2*; *Figure 6A*). These distinct sensory cell clusters project to distinct neuropils in the centre of the brain (*Figure 6A–C*). A few sensory cell types are more scattered in the ventro-median head (*Figure 6F*). The interneuron somata form clusters in the mushroom bodies, in the eye and dorso-lateral clusters, the apical organ, and in the ventro-median domain of the brain (*Figure 6D*). The interneuron projections are also organised into overlapping neuropils (*Figure 6E*; *Video 2*).

The head contains 45 sensory neuron types with postsynaptic partners, including 52 head interneuron types directly postsynaptic to sensory neurons (*Figure 3*).

The cell-type annotation and connectivity information for all differentiated neurons allowed us to investigate the cellular composition and circuitry of the different head ganglia and cell types.

Information flows from various sensory neuron clusters to head interneurons that converge on ciliomotor, head ventral motoneuron (vMNs), and head descending interneuron groups (*Figure 6H*). There is recurrent connectivity between some

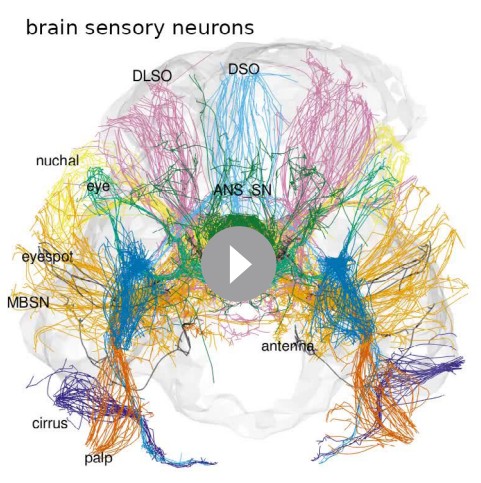

**Video 2.** Organisation of brain neuropils and ganglia in the 3-day-old larva. Morphological renderings of sensory, inter- and motoneurons are shown separately. Anterior view.

https://elifesciences.org/articles/97964/figures#video2

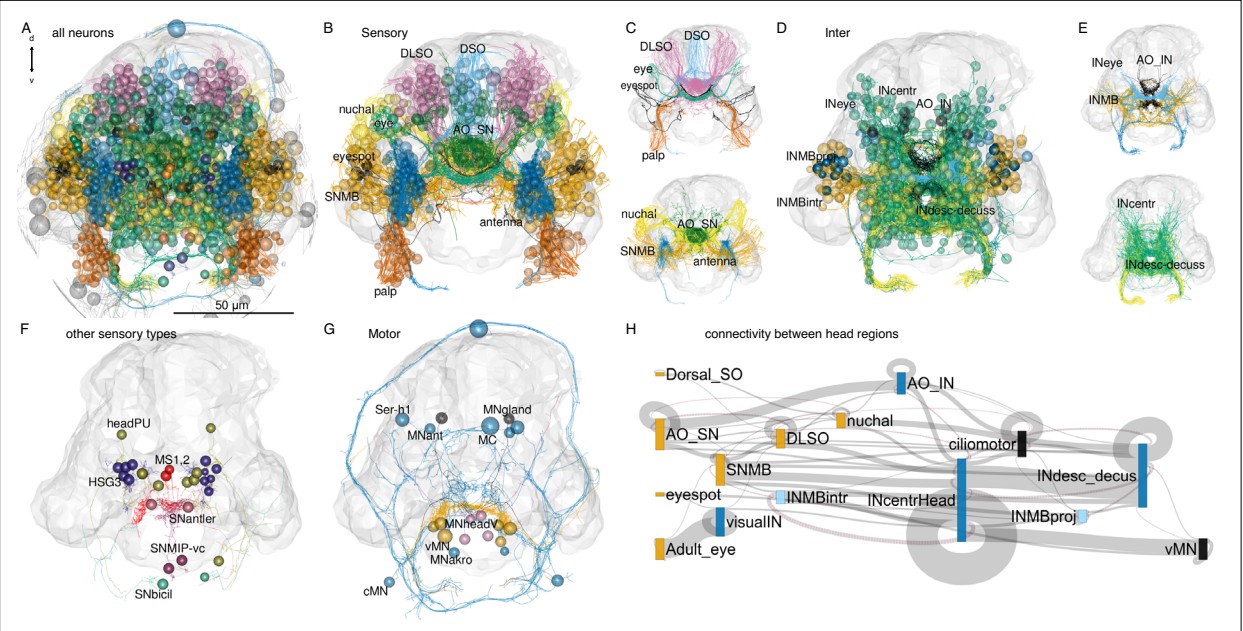

**Figure 6.** Anatomy of head neuropils and global organisation of brain connectivity. (**A**) Morphological rendering of all head neurons. (**B**) Head sensory neurons coloured by head sensory ganglia. Abbreviations: DLSO, dorso-lateral sense organ; DSO, dorsal sense organ; SNMB, mushroom body sensory neuron; INMB, mushroom body interneuron; INMBintr, mushroom-body-intrinsic interneuron; AO SN, apical organ sensory neuron; AO IN, apical organ interneuron; INcentr, central brain interneuron; vMN, ventral motoneuron. (**C**) Same cells as in (**B**) rendered without the cell soma to show neuropil organisation. (**D**) Head interneurons coloured by head ganglia. (**E**) Same cells as in (**D**) rendered without the cell soma to show neuropil organisation. (**F**) Rendering of other sensory cell types that do not form separate neuropils. (**G**) Rendering of head motoneurons. (**H**) Sankey circuit diagram showing information flow (from left to right) based on synaptic connectivity between cell categories of head ganglia. Bars represent groups of neurons, grey connecting lines represent synaptic connections (pre-to-post organised left-to-right). Magenta lines represent right-to-left connections. Only connections with >10 synapses are shown. *Figure 6—source data 1*. Connectivity matrix of the network in panel **H**.

The online version of this article includes the following source data and figure supplement(s) for figure 6:

**Source data 1.** Source data of the network in panel H.

**Figure supplement 1.** Head cell types and circuits.

interneuron clusters (e.g. between INMBintr and INcentrHead) and within-cluster connectivity (e.g. visualIN, INcentralHead).

One direct motor output of head circuits are three pairs of ventral head motoneurons (the vMNs: MN1, MN2, MN3) (*Randel et al., 2015*; *Randel et al., 2014*). These decussating (crossing the midline before descending) motoneurons are at the core of a postural control module (*Figure 2*; *Figure 7A–D*). Their inputs include central head interneurons and several premotor sensory neurons (SNantler, MS, eyespotPRCR3) that directly synapse on the vMNs. The vMNs have a decussating morphology and innervate contralateral ciliated cells and ventral and dorsal longitudinal muscles, controlling trunk bending and the laterality of ciliary beating (*Randel et al., 2015*; *Randel et al., 2014*).

Another motor output from the head is two types of exocrine glands in the first segment (ciliated gland, MVgland). These are innervated by two MNgland-head gland-motor neurons. The MNgland-head cells receive input from the rhythmically active serotonergic Ser-h1 ciliomotor neurons (*Verasztó et al., 2017*), suggesting a link between ciliary activity and glandular secretion.

Furthermore, several head ciliomotor neurons directly innervate locomotor ciliary bands (*Verasztó et al., 2017*) (MC, Ser-h1, MNant, vMN). Of these, the MNant cells receive direct input from several head sensory neurons (SNhook, SNtorii, SNantler, SNhorn) (*Figure 7F*). The vMN and MNant motoneurons thus seem to integrate multiple direct sensory and other inputs.

Sensory integration also characterises the visual system. Here, the visual eye photoreceptors (PRC) share a postsynaptic interneuron (INR) with the PRCR1 photoreceptors of the eyespots (*Figure 7G and H*).

To identify neurons with possible integrative function, we ranked cells by the number of presynaptic and postsynaptic cell-type-level partners (*Figure 7J and K*). We also ranked cells by various

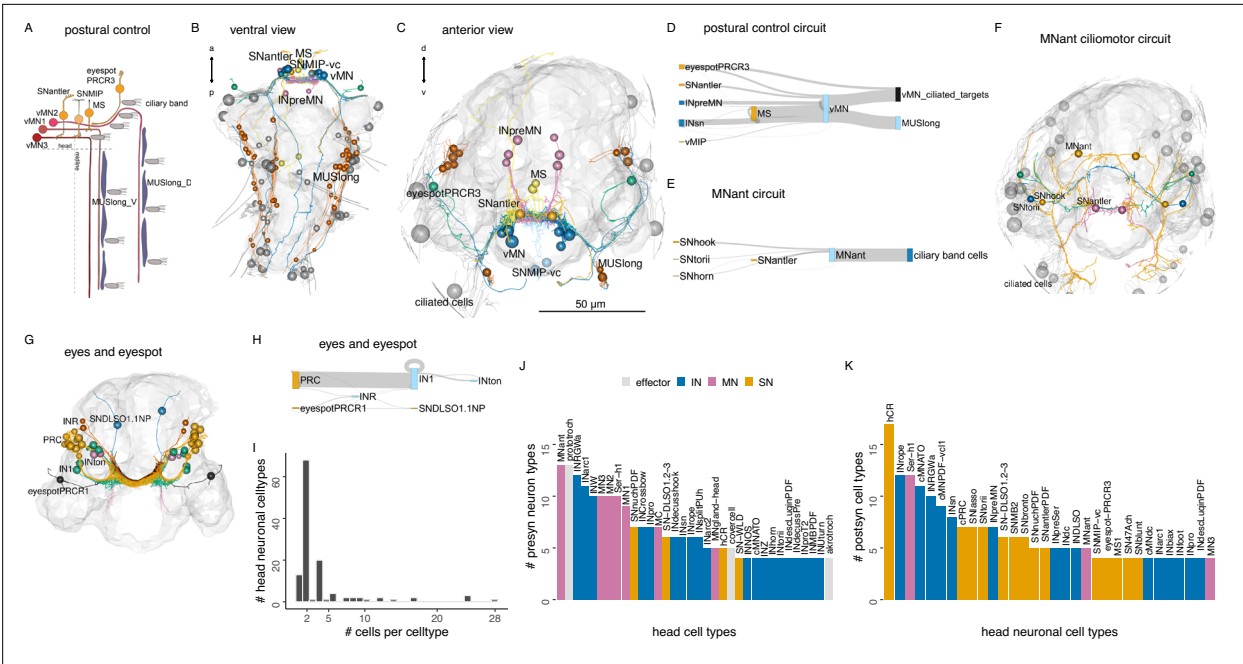

**Figure 7.** Brain circuits and cell types. (**A**) Schematic diagram of the postural control system of vMN neurons with their direct sensory neuron inputs and motor outputs. (**B, C**) Morphological rendering of neurons and effector cells of the postural control system, ventral (**B**) and anterior (**C**) views. (**D**) Sankey diagram of synaptic connectivity in the postural control system. (**E**) Sankey diagram of synaptic connectivity of MNant motoneurons. (**F**) Morphological rendering of the MNant ciliomotor circuit, anterior view. (**G**) Morphological rendering of eyespot and visual eye photoreceptors and their direct postsynaptic partners. (**H**) Sankey diagram of synaptic connectivity of the eyespot and visual eyes. (**I**) Histogram of the number of cells per head neuronal cell type. (**J**) Number of presynaptic neuron types for different head cell types (at least four synapses). (**K**) Number of postsynaptic cell types for different head neuronal cell types (>3 synapses). In **B**, **C**, **F**, and **G** the outline of the yolk is shown for reference. *Figure 7—source data 1*. Connectivity matrix of the network in panel D. *Figure 7—source data 2*. Connectivity matrix of the network in panel E. *Figure 7—source data 3*. Connectivity matrix of the network in panel H.

The online version of this article includes the following source data and figure supplement(s) for figure 7:

**Source data 1.** Connectivity matrix of the network in panel D.

**Source data 2.** Connectivity matrix of the network in panel E.

**Source data 3.** Connectivity matrix of the network in panel H.

**Figure supplement 1.** Gland cells and their innervation.

**Figure supplement 1—source data 1.** Source data for MNgland-head connectivity.

**Figure supplement 2.** Metrics of head cell types.

**Figure supplement 3.** Number of partners of head sensory and interneuron types.

node-centrality measures, including node weighted degree, PageRank, betweenness, and authority (*Figure 7—figure supplement 2*). PageRank is an algorithm used by Google to rank webpages and scores the number and quality of the incoming links of a node (*Page et al., 1999*), betweenness centrality measures the number of shortest paths that pass through a node in a graph (*Freeman, 1977*), and authority measures the extent of inputs to a node by hubs in a network (*Kleinberg, 1999*).

The MNant and vMN motoneurons have among the highest number of presynaptic partners. Among the interneurons, INRGW, INarc1, and INW integrate the largest number of inputs.

Regarding the multiplicity of outputs, the head collar receptor neurons (hCR) that mediate a startle response (*Bezares-Calderón et al., 2018*) stand out with the largest number of postsynaptic cell types.

Some sensory neurons directly connect to muscle-motor, ciliomotor, or pigment-motor neurons (e.g. eyespotPRC_R3, MS cells, SNhook, SNblunt) (*Figure 6—figure supplement 1*, *Figure 7*) or to effector cells (e.g. eyespotPRC_R3, hPU2l_asymPDF) (*Figure 7*, *Table 1*).

Analysis of connectivity (>2 synapses) only between head sensory and interneurons revealed a skewed distribution with many sensory neurons only synapsing on one interneuron type and many

interneurons only receiving input from one sensory neuron type (*Figure 7—figure supplement 2A and B*).

At the other end of the distribution, the head CR neurons have eight and SNlasso have five types of postsynaptic interneuron partners, suggesting that these cells recruit extended downstream circuits (*Figure 7—figure supplement 3A and E*).

Some interneurons and motoneurons are directly postsynaptic to several distinct sensory neuron types (up to 9, >2 synapses) (*Figure 7—figure supplement 3*). Among the interneurons, INRGWa, INarc1, and INcrossbow have >4 presynaptic sensory neuron partners. The MNant and ventral head motoneurons (vMN) receive direct input from 2 to 5 sensory neuron types (*Figure 7*, *Figure 7—figure supplements 2 and 3*).

The interneurons with the highest number of presynaptic interneuron partners are the INW cells (*Figure 7—figure supplement 3D*). INWs are projection neurons of the central brain with decussating axons that delineate a V-shaped brain neuropil (Figure 9P; *Figure 7—figure supplement 3H*). They receive inputs from the mushroom body and central brain (see below). The interneurons with the largest number of postsynaptic interneuron partners are the INRGWa cells of the anterior neurosecretory nervous system (*Figure 7—figure supplement 3F and G*).

## Mushroom-body anatomy and circuits

Nereid annelids have a pair of mushroom bodies with morphological and molecular similarities to arthropod mushroom bodies (*Tomer et al., 2010*). In *Platynereis*, the mushroom bodies start to develop in the larva. In the 6-day-old volume, the partial reconstruction of neurite projections revealed the presence of both sensory and interneurons in the mushroom bodies. Some of the sensory neurons were found to project to the anterior neurosecretory neuropil (*Arendt et al., 2021*; *Vergara et al., 2021*). Here, we present the complete reconstruction of the developing mushroom bodies and their circuits in the 3-day-old larva (*Video 3*; *Figure 8*).

The mushroom bodies (MBs) are formed by a pair of ventrolateral brain ganglia. They comprise both sensory and interneurons (SNMB and INMB) (*Figure 8A–D*). Most SNMB cells project to two lateral MB neuropils and a few to the anterior neurosecretory plexus (*Figure 8B*), in agreement with the projection reconstructions (*Vergara et al., 2021*).

We classified INMBs into intrinsic (MBintrIN) and output projection (MBON) types (*Figure 8C and D*). MBintrINs project to the MB neuropils, MBONs project ventrally to the VNC or to the developing mouth (stomodeum) (*Figure 8C–G*; *Video 3*).

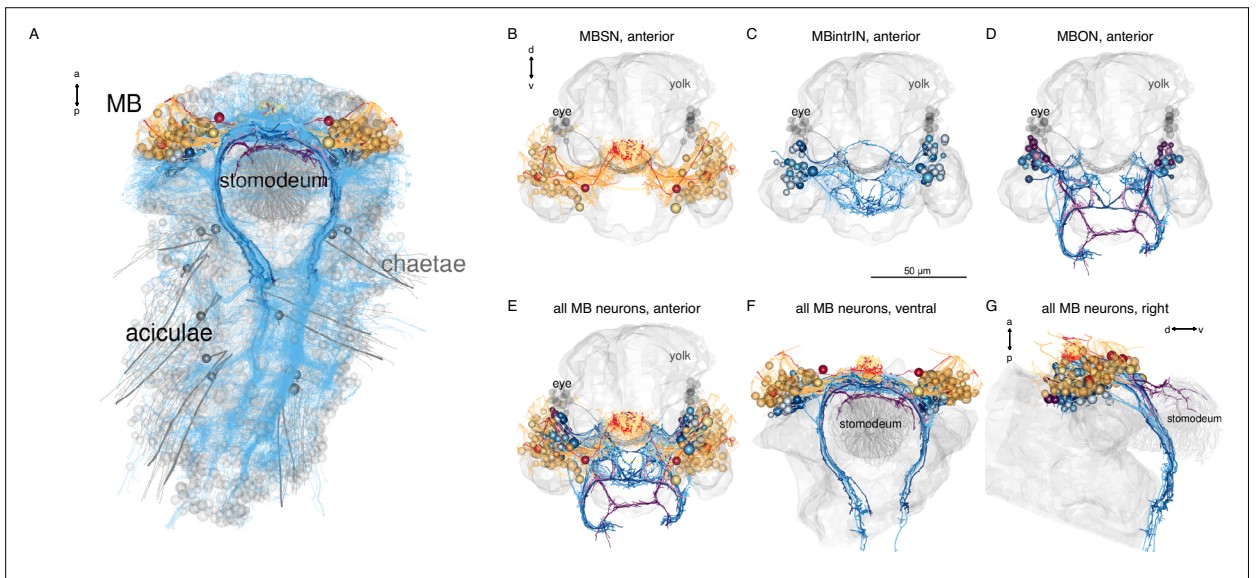

**Figure 8.** Anatomy of the *Platynereis* mushroom bodies. (**A**) Morphological rendering of the *Platynereis* mushroom bodies in the context of the entire nervous system (neurites shown in cyan), ventral view. (**B–G**) Morphological renderings of all mushroom body sensory neurons (**B**), intrinsic interneurons (**C**), output neurons (**D**), and all neurons in three different views (**E–G**). The visual eyes, yolk outline, and the stomodeum are shown in grey for reference.

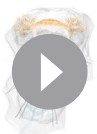

**Video 3.** Organisation and cell-type composition of the mushroom bodies in the 3-day-old larva, ventral and anterior views.
https://elifesciences.org/articles/97964/figures#video3

Besides MB-intrinsic sensory neurons, MBs receive sensory input from the antennae, the palps, and two further sensory neuron types (SNbronto, SNstiff) (*Figure 9A–D*). These sensory inputs are mostly to the MBONs. SNMBs connect to MBINs, central brain interneurons, and the MNant ciliomotor neurons (*Figure 9A*). There is cell-type-specific connectivity between sensory neuron types and interneurons, with groups organised into specific micro-circuits (*Figure 9B, E–G*, *Figure 9—figure supplement 1A and B*). MBINs connect to distinct central brain interneuron or projection-neuron types that in turn provide distinct inputs to trunk circuits (*Figure 9H–Q*, *Figure 9—figure supplement 1C and D*), including motor innervation (*Figure 9O*).

The morphology and connectivity of distinct mushroom body cell types shows left-right symmetry. The Sholl diagrams are similar for left-right pairs and the correlation coefficient of the left and right connectivity matrices is 0.76 (*Figure 9—figure supplement 2*).

The outputs of the mushroom body circuits also include projection neurons of the central brain, including the INW, INlasso, INproT2, and INdecussHook neurons (*Figure 9H–Q*). Among these, the INproT2 neurons are premotor neurons synapsing on trunk MNring motoneurons, indicating a shallow sensory-motor organisation of mushroom-body outputs.

The overall architecture of the *Platynereis* MB is multilayer, parallel, and feed-forward, with only very weak recurrent connections or lateral connections between the feed-forward networks (*Figure 9B*; *Figure 9—figure supplement 1*). The circuit also shows a fan-out fan-in architecture, with the sensory neurons diverging to several IN targets (e.g. SNtorii to INtorii, INbigloop, and INhorn) and post-MB interneurons receiving converging input (e.g. INW from INhook, INlasso, INMBtype5-6, and SNlasso). This multilayer perceptron architecture may support the formation of associative memories, as proposed for memory circuits in cephalopods (*Shomrat et al., 2011*).

## Head-to-trunk and left-right connectivity

The whole-body resource allowed us to examine all connections linking the head and the trunk (*Figure 10A and B*).

There are 138 brain neurons that descend to the VNC and 75 VNC neurons that ascend to the brain (*Figure 10C and D*). There are 549 trunk connectome cells that receive synapses from head neurons and 288 head connectome cells that receive synapses from trunk neurons (*Figure 10E and G*).

The trunk-projecting brain neurons have either a decussating or ipsilaterally descending morphology. The two types project to different domains of the VNC and have different target neurons (*Figure 10—figure supplement 1E and F*). Decussating and descending neurons receive most of their synaptic inputs (postsynaptic sites) in the central head neuropil and form most of their presynapses in the VNC (*Figure 10—figure supplement 1C, D, H–M*).

Ranking head neurons by the number of synapses on trunk targets highlights the strong motor connections of the INrope MB projection neurons (*Bezares-Calderón et al., 2018*) and the ventral head motoneurons (vMNs) (*Figure 10F*). The MNant and MNgland-head neurons also innervate trunk effectors (ciliary band and gland cells, respectively). The other head projection neurons predominantly connect to trunk interneurons (*Figure 10F*).

Trunk targets of head neurons can be distributed across segments 1–3 (INrope, MN1, MN2) and also reach the most posterior pygidium (MN3) or occur mostly in segment 1 (most head SN and INs with trunk connections)(*Figure 10—figure supplement 2*).

The strongest inputs from trunk neurons to the head are provided by the cioMNcover, pygPBunp, and MC3cover pigment-motor neurons and the Loop and Ser-tr1 ciliomotor neurons. Other contacts are formed mostly between trunk and head interneurons (*Figure 10H*).

Plotting head-to-trunk connectivity by cell class (sensory, inter, motor neurons, and effectors) (*Figure 10—figure supplement 3*) shows the relative strengths of the connections.

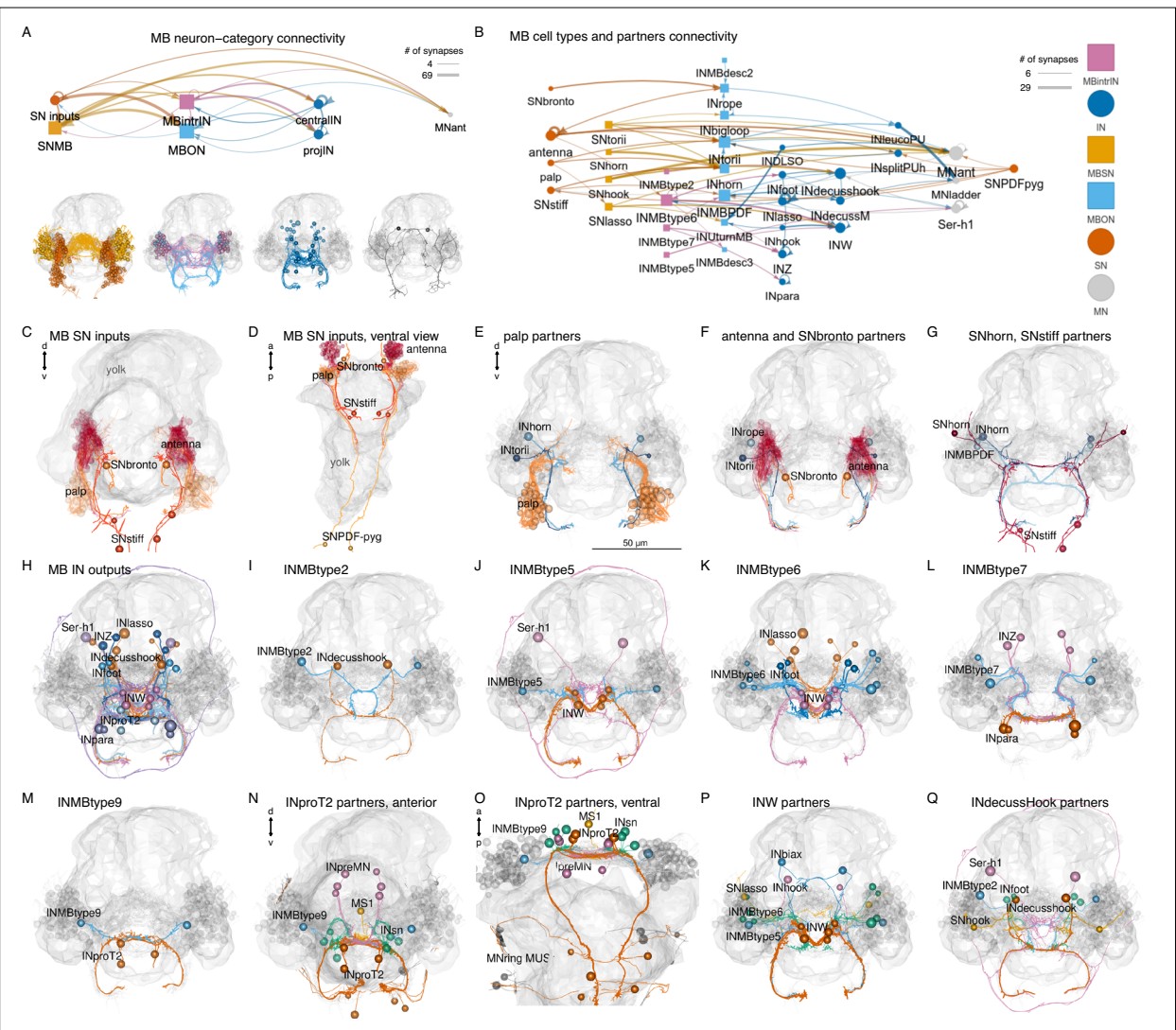

**Figure 9.** Parallel circuit organisation of the mushroom bodies. (**A**) Connectivity of the mushroom body by neuron category with morphological rendering for each category shown below. (**B**) Connectivity of mushroom-body neurons and their outputs. Nodes represent neurons grouped by cell type, arrows show synaptic connectivity. (**C, D**) Sensory inputs to the mushroom bodies (other than SNMB), anterior (**C**) and ventral (**D**) views. (**E**) Palp sensory neurons and their interneuron targets in the mushroom body. (**F**) Antennal and SNbronto sensory neurons and their interneuron targets in the mushroom body. (**G**) SNhorn and SNstiff sensory neurons and their interneuron targets in the mushroom body. (**H**) Interneuron inputs to the mushroom body. (**I–L**) Inputs to INMBtype2 (**I**), INMBtype5 (**J**), INMBtype6 (**K**), and INMBtype7 (**L**) mushroom body interneurons. (**M**) INMBtype9 and its postsynaptic target, the INproT2 premotor projection neuron. (**N, O**) Pre- and postsynaptic partners of INproT2, anterior (**N**) and ventral (**O**) views. (**P**) Partner of the INW projection neurons. (**Q**) Partners of the INdecusshook projection neurons. Yolk outline and all mushroom body neurons are shown in grey for reference. Abbreviations: SN, sensory neuron; IN, interneuron; MN, motor neuron; MBSN, mushroom body sensory neuron; MBintrIN, mushroom body intrinsic interneuron; MBON, mushroom body output neuron. *Figure 9—source data 1*. Connectivity matrix of the network in panel **A**. *Figure 9—source data 2*. Connectivity matrix of the network in panel **B**.

The online version of this article includes the following source data and figure supplement(s) for figure 9:

**Source data 1.** Connectivity matrix of the network in panel A.

**Source data 2.** Connectivity matrix of the network in panel B.

**Figure supplement 1.** The mushroom body circuit with different subcircuits highlighted.

**Figure supplement 2.** Left-right symmetry of mushroom body cell types and their wiring.

**Figure supplement 2—source data 1.** Source data for panel A with average Sholl data for mushroom body cell types on the left side.

**Figure supplement 2—source data 2.** Source data for panel B with average Sholl data for mushroom body cell types on the right side.

**Figure supplement 2—source data 3.** Connectivity matrix of mushroom body cell types on the left side.

**Figure supplement 2—source data 4.** Connectivity matrix of mushroom body cell types on the right side.

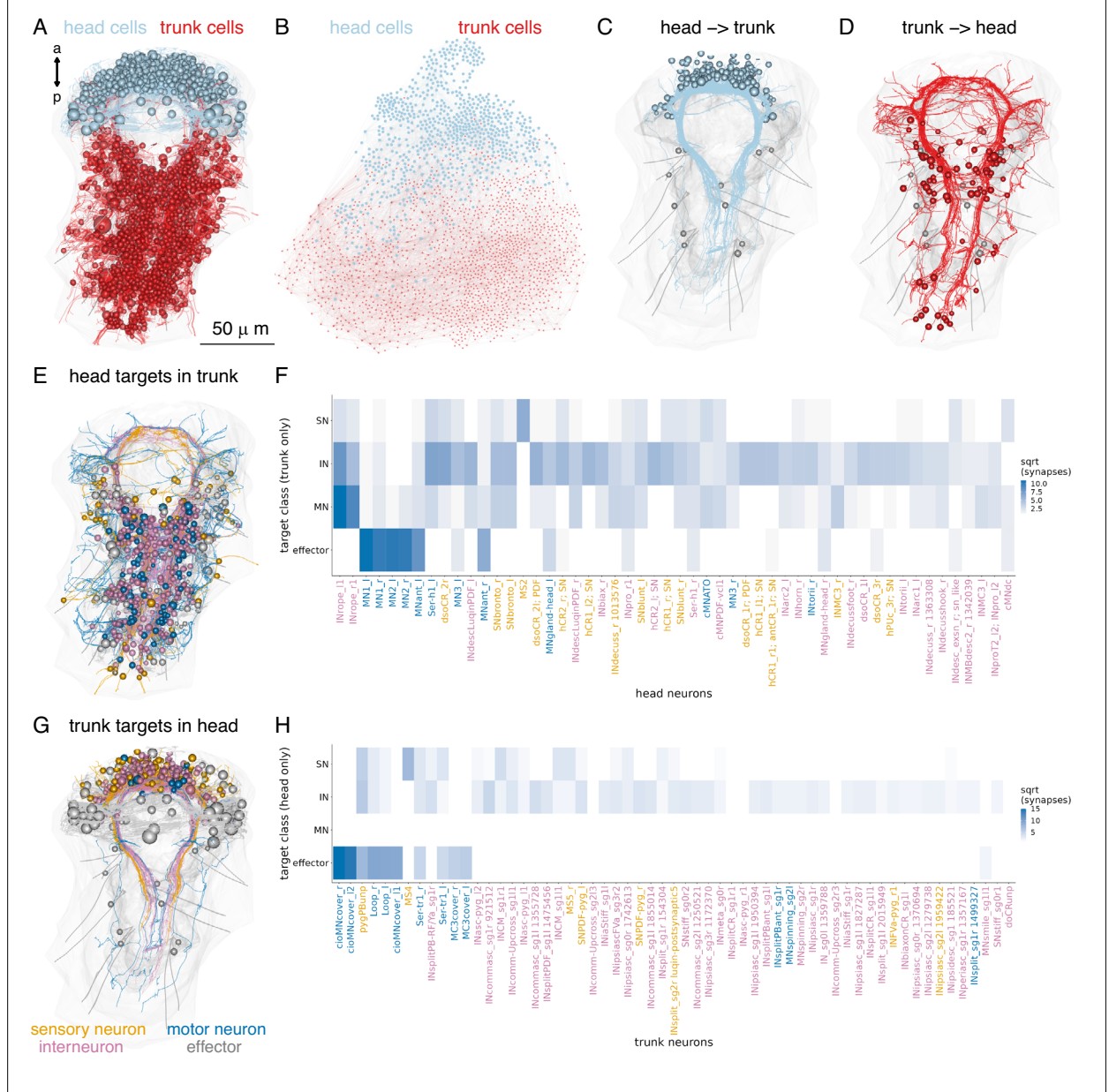

**Figure 10.** Connections between the head and the trunk. (**A**) Morphological rendering of head (cyan) and trunk (red) cells, which are part of the connectome. (**B**) Connectome graph with head (cyan) and trunk (red) cells coloured separately. (**C**) Morphological rendering of all head neurons with descending projections into the ventral nerve cord. (**D**) Morphological rendering of all trunk neurons with ascending projections into the head. (**E**) Morphological rendering of all synaptic targets of head neurons in the trunk coloured by cell class. (**F**) Distribution across classes of trunk targets of head neurons ordered by the number of head to trunk synapses (top 50 neurons shown). (**G**) Morphological rendering of all synaptic targets of trunk neurons in the head coloured by cell class. (**H**) Distribution across classes of head targets of trunk neurons ordered by the number of trunk to head synapses (top 50 neurons shown). *Figure 10—source data 1*. Source data for panel **F**. *Figure 10—source data 2*. Source data for panel **H**.

The online version of this article includes the following source data and figure supplement(s) for figure 10:

**Source data 1.** Source data for panel F showing the distribution across classes of trunk targets of head neurons.

**Source data 2.** Source data for panel H showing the distribution across classes of head targets of trunk neurons.

**Figure supplement 1.** Head descending and decussating neurons.

**Figure supplement 1—source data 1.** Source data for panels (J, M) with mean radial density of post- and presynapses of head descending and decussating neurons.

**Figure supplement 2.** Head-trunk connectivity.

*Figure 10 continued on next page*

*Figure 10 continued*

**Figure supplement 3.** Connectivity of head-trunk cell groups.

**Figure supplement 3—source data 1.** The source data matrix of connectivitty between head-trunk cell groups as a csv file.

**Figure supplement 4.** Connectivity of left-right cell groups.

**Figure supplement 4—source data 1.** The source data matrix with the connectivity of left-right cell groups saved as a csv file.

A similar analysis for left-to-right cell groups highlights the strong connectivity between the two body sides across all cell classes (*Figure 10—figure supplement 4*). For example, left-side eyespot-PRCR3, SNantlerPDF, and SNMIP-vc sensory neurons synapse on right-side MN1 and MN3 motoneurons, INrope interneurons synapse on both left and right MNspinning mononeurons, and mononeurons form a similar number of synapses on left- and right-side partners (*Figure 10—figure supplement 4*).

## Intersegmental connectivity

Next, we analysed the neuronal complement and interconnectivity of the body segments.

The 3-day-old larva has three main trunk segments with chaeta-bearing parapodia (chaetigerous segments) and a more anterior cryptic segment (*Saudemont et al., 2008*; *Steinmetz et al., 2011*). In addition, the pygidium forms the posterior-most part of the body (*Starunov et al., 2015*; *Figures 1A and 11A*). The ciliary bands mark the posterior segment boundaries.

Breaking down connections to segments (head, sg0-3, pygidium) or to cell classes (SN, IN, MN, effector) and segments shows that the strongest connections are formed within segments, but there are also connections between any pair of segments with the exception of segment 0 (*Figure 11B–D, F and G*; *Figure 11—figure supplement 1*). Segment 1 interneurons, for example, connect to cells in four other body regions (*Figure 11G*). In agreement with this, morphological rendering of cells per segment reveals long-range projections beyond the boundary of each segment (*Figure 11H–K*). The head, segment 1, and the pygidium all contain neurons that project along the entire length of the body (neurons with 'global reach')(*Figure 11L–N*).

The innervation of different effector classes shows segment-selectivity. Ciliary band cells receive most of their innervation from the head and first segment (*Figure 11E*). Gland cells receive most of their synapses from the second segment. Muscles are innervated by motoneurons with somas in the head and segments 1–3. In contrast, pigment cells receive their inputs predominantly from pygidial neurons (*Figure 11E*).

Overall, intersegmental connectivity is a hallmark of the *Platynereis* larval nervous system and suggests that motor control and behaviour cannot be understood by focusing on a single segment or the brain alone.

## Segment-specific cell types and circuits in the trunk

We identified many trunk-specific and segment-specific neuron types and neuron types present in different subsets of segments (*Figure 12*, *Figure 12—figure supplements 1–3*), revealing a heteromeric organisation of the annelid larval trunk. There are 10 neuronal cell types specific to the first segment, eight to the second segment, two to the third segment, and seven to the pygidium (*Figure 12*, *Figure 12—figure supplement 3*).

These distinct sets of neurons suggest a functional specialisation of the different trunk segments and are in agreement with segmental differences in the expression of developmental transcription factors (*Vergara et al., 2017*).

The segment-specific cell types form unique motor circuits. The first segment contains ciliomotor neurons involved in body-wide ciliary closures and beating (*Figure 12*; *Verasztó et al., 2017*).

In addition, segment-1-specific inter- and motoneurons have extensive connections with other trunk and head cell types (*Figure 12D*), including sensory inputs and motor outputs.

Segment 2 contains a unique pair of giant motoneurons (MNspinning) that extensively innervate the contralateral spinning glands (spinGland) in the second and third segments (*Figure 12E and F*). The four spinGland cells have a large microvillar secretory pore at the tip of the ventral parapodia (neuropodia). MNspinning cells form reciprocal connections with pseudounipolar INsplit mechanosensory interneurons (*Figure 12I*). Segment 2 also contains the large contra- and ipsilaterally projecting MNbow motoneurons and four types of unique interneurons (*Figure 12G and H*).

The pygidium (*Figure 12J*) contains several unique cell types with global reach (*Figure 12—figure supplement 3*). The cioMNcover and pygPBunp motoneurons innervate the head cover cells (with large pigment vacuoles) that occur in two rings anterior and posterior to the head protoroch ciliary band (*Figure 12J and K*). The pygPBunp neuron also synapses on another head pigmented cell type (vacuolar cell). Pygidial cells also form connections with segment 1–3 sensory, inter-, and motoneurons.

Ranking trunk and pygidial cell types by various network centrality measures often identified segment-specific cells as strongly connected (e.g. pygPBunp, Ser-tr1, Loop)(*Figure 12—figure supplement 2*).

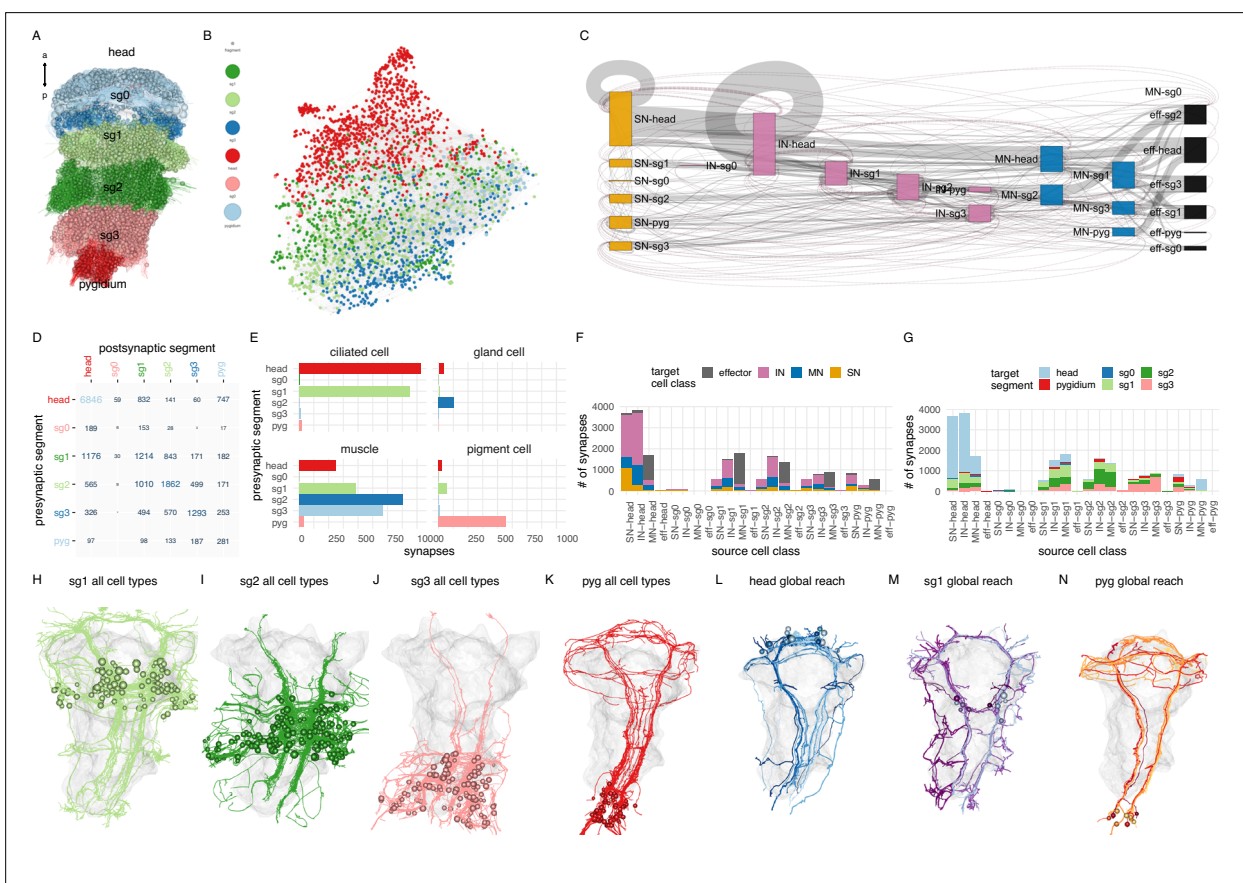

**Figure 11.** Intersegmental connectivity in the larva. (**A**) All cells in the body, coloured by body region. (**B**) Connectome graph with nodes coloured by segment. (**C**) Sankey connectivity diagram of sensory (orange), interneurons (cyan), motoneurons (blue), and effectors (purple) in the different body regions. Edge thickness is proportional to the number of synaptic connections. (**D**) Number of synaptic connections linking the six body regions. (**E**) (**F**) Distribution of synapses across different target cell classes for every source cell class (SN, IN, MN, grouped by body region). (**G**) Distribution of synapses across target body regions for every source cell class (SN, IN, MN, grouped by body region). (**H–K**) Morphological rendering of all neuronal cell types in segments 1–3 and the pygidium showing cross-segmental neurite projections. (**L–N**) Neurons with global (whole-body) projections in the head (**L**), first segment (**M**), and the pygidium (**N**). In (**H–N**), the yolk outline is shown in gray for reference. *Figure 11—source data 1*. Source data for panel C. *Figure 11—source data 2*. Source data for panel D. *Figure 11—source data 3*. Source data for panel E. *Figure 11—source data 4*. Source data for panels **F** and **G**.

The online version of this article includes the following source data and figure supplement(s) for figure 11:

**Source data 1.** Source data for panel C with connectivity data of sensory neurons, interneurons, motoneurons and effectors in the different body regions.

**Source data 2.** Source data for panel D with connectivity data between the six body regions.

**Source data 3.** Source data for panel E with synapse data across different target cell classes for every source cell class.

**Source data 4.** Source data for panels F and G.

**Figure supplement 1.** Connectivity matrix of cell categories across the six body regions.

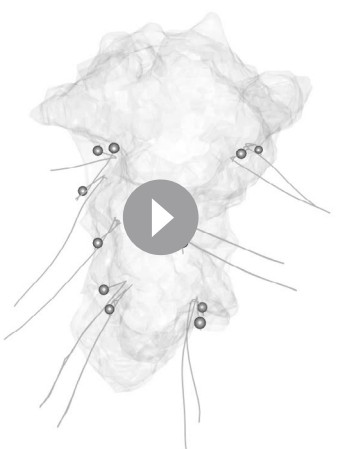

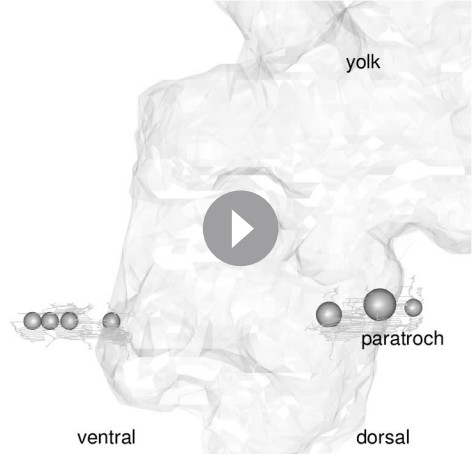

**Video 4.** Morphological rendering of sensory, inter- and motoneurons of the mechanosensory girdle in the 3-day-old larva, ventral and anterior views.

https://elifesciences.org/articles/97964/figures#video4

**Video 5.** Morphological rendering of muscles (grey) and sensory cell types in the periferal nervous system of the 3-day-old larva. Only cells in the second segment on the left side are shown.

https://elifesciences.org/articles/97964/figures#video5

## Serially repeated cells across the body segments

The whole-body connectome allowed us to systematically investigate the occurrence of serially repeated cell types or cell-type families in the annelid body.

Each larval segment has a distinct neuron-type composition with some unique cell types (*Figure 12*). At the same time, several neuron types occur in all chaetigerous segments. These include the chaeMech sensory neurons (Figure 15A), the INsplitPB (Figure 16B), and INbackcross interneurons and the MNche and MNhose motoneurons (not shown).

Four cell-type families (CR, PU, PB, INsplit) and two muscle types (MUSlongV, MUStrans) occur across five or six body regions (head, segments 0–3, and pygidium) (*Figure 13*). This pattern may indicate serial homology, suggesting a metameric evolutionary origin for the trunk segments, the pygidium, and the head.

## The mechanosensory girdle

The mechanosensory neurons and their postsynaptic partners form an anatomically distinct system that spans the entire body of the larva. Based on its morphology with a circumoral ring that continues in two VNC tracts, we termed this system the mechanosensory girdle (*Figure 14*; *Figure 14—figure supplement 1*; *Video 4*).

The girdle includes the penetrating vibration-sensing collar receptor (CR) neurons (*Bezares-Calderón et al., 2018*), the penetrating biciliated (PB) neurons, the penetrating uniciliated (PU) neurons, the interparapodial penetrating multiciliated (interparaPM) neurons (*Video 5*). These penetrating cells all have one or more penetrating sensory cilia surrounded by a collar of microvilli. The CR and PB neurons express the mechanosensory polycystin PKD2-1, with CRs also expressing PKD1-1 (*Bezares-Calderón et al., 2018*). The girdle also includes the chaeMech dendritic cells that are chaetal mechanoreceptors (*Figure 15* and see below).

The global projections of two mechanosensory interneuron types — INsplitPBant and INsplitPB-RF/Ya — outline the axon tracts of the mechanosensory girdle (*Video 4*; *Figure 14—figure supplement 1*).

The projections of several other neurons follow this axonal tract and form a bundle at the ventral side of the VNC (*Figure 14—figure supplement 2*).

The analysis of axon tracts and connectivity allowed us to identify further sensory cell types that are part of the mechanosensory girdle. These include the head SNbronto (dendritic), SNblunt (blunt sensory ending beneath the cuticle, no cilium), and the pygidial SNPDF-pyg and SNpygM cell types (*Figure 14—figure supplement 1*).

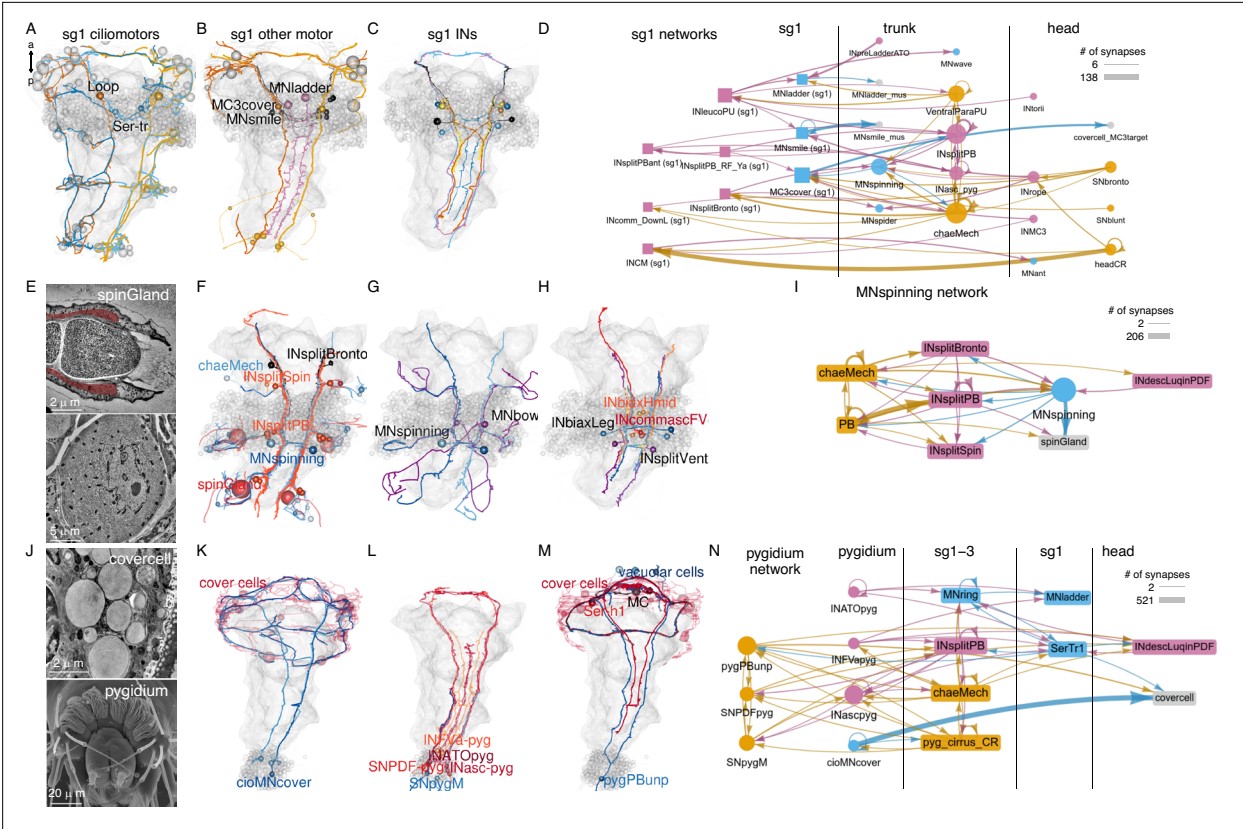

**Figure 12.** Segment-specific circuits. (**A**) Morphological rendering of segment-1-specific ciliomotor neurons. (**B**) Other segment-1-specific motoneurons. (**C**) Segment-1-specific interneurons. (**D**) Grouped connectivity graph of segment-1-specific cell types and their synaptic partners in other trunk segments and the head. (**E**) Transmission electron micrograph (TEM) of the spinGland nozzle and a secretory vesicle. (**F**) The MNspinning motoneurons with their spinGland targets and presynaptic partners. (**G**) The segment-2-specific MNspining and MNbox motoneurons. (**H**) Segment-2-specific interneurons. (**I**) Grouped connectivity graph of segment-2-specific cell types and their synaptic partners. (**J**) TEM image of a cover cell covering the prototroch ciliary band (top) and SEM image of the pygidium. (**K**) The pygidium-specific cioMNcover cells and their cover cell targets. (**L**) Pygidium-specific sensory and interneurons. (**M**) The pygidium-specific pygPBunp sensory neuron and its targets in the head. (**N**) Grouped connectivity graph of pygidium-specific neurons and their synaptic partners in the trunk and head. *Figure 12—source data 1*. Connectivity matrix for the network in panel **D**. *Figure 12—source data 2*. Connectivity matrix for the network in panel **I**. *Figure 12—source data 3*. Connectivity matrix for the network in panel **N**.

The online version of this article includes the following source data and figure supplement(s) for figure 12:

**Source data 1.** Connectivity matrix for the network in panel D with grouped connectivity data of segment-1-specific cell types and their synaptic partners in other trunk segments and the head.

**Source data 2.** Connectivity matrix for the network in panel I with grouped connectivity data of segment-2-specific cell types and their synaptic partners.

**Source data 3.** Connectivity matrix for the network in panel N with grouped connectivity data of pygidium-specific neurons and their synaptic partners in the trunk and head.

**Figure supplement 1.** Statistics of trunk cell types.

**Figure supplement 2.** Partners and centrality measures of trunk cell types.

**Figure supplement 3.** Segment-specific cell types.

The interneurons in the girdle include various INsplit cells (see below) as well as the pygidial ascending INasc-pyg and the head descending INrope neurons (*Bezares-Calderón et al., 2018*; *Figure 14—figure supplement 1*).

The ciliomotor neurons MNant (*Verasztó et al., 2017*; *Figure 6G*) and the cover-cell motoneurons MC3cover (*Figure 12B*) are also part of the girdle.

Overall, the anatomy and connectivity suggest that the girdle forms a separate VNC tract for the processing of mechanosensory signals (*Figure 14—figure supplement 2*).

## Chaetal mechanoreceptors

The trunk neurons with the largest number of postsynaptic partners and the highest weighted degree in the grouped connectome are the chaeMech chaetal mechanoreceptor neurons (*Figure 12—figure supplement 1*).

The chaeMechs are part of the mechanosensory girdle and are dendritic sensory cells in segments 1–3 in both the dorsal (notopodium) and the ventral (neuropodium) lobes of the parapodia (*Video 5*). The sensory dendrites of chaeMech cells branch between the chaetal sacs (*Figure 15A–D*) and may sense the displacement of the chaetae during crawling (proprioception) or due to external mechanical stimuli. The annelids *Harmothoë* (a polynoid) and *Nereis* (a nereid) have cells with a similar morphology called bristle receptors. These cells show rapidly adapting spikes upon the displacement of the chaetae (*Dorsett, 1964*; *Horridge, 1963*). The chaeMech cells are also reminiscent of dendritic proprioceptors in *Drosophila* larvae that sense body-wall deformations and provide feedback about body position through premotor neurons (*He et al., 2019*; *Vaadia et al., 2019*; *Zarin et al., 2019*). In *Harmothoë* and *Nereis*, the bristle receptors connect to the giant axon system. In *Platynereis*, chaeMech neurons are highly interconnected. They project into the VNC where inputs and outputs are mixed on the chaeMech axons (*Figure 15B*).

The direct postsynaptic partners of chaeMech cells include the premotor interneurons INsplitCR, INsplitBronto, INchaeMech, and the MNring and MNspinning motoneurons (*Figure 15E and F*). The innervation of MNspinning by chaeMech (and also indirectly by PB mechanosensory neurons; *Figure 12I*) suggests that parapodial mechanosensation may regulate glandular secretion of the spinGlands.

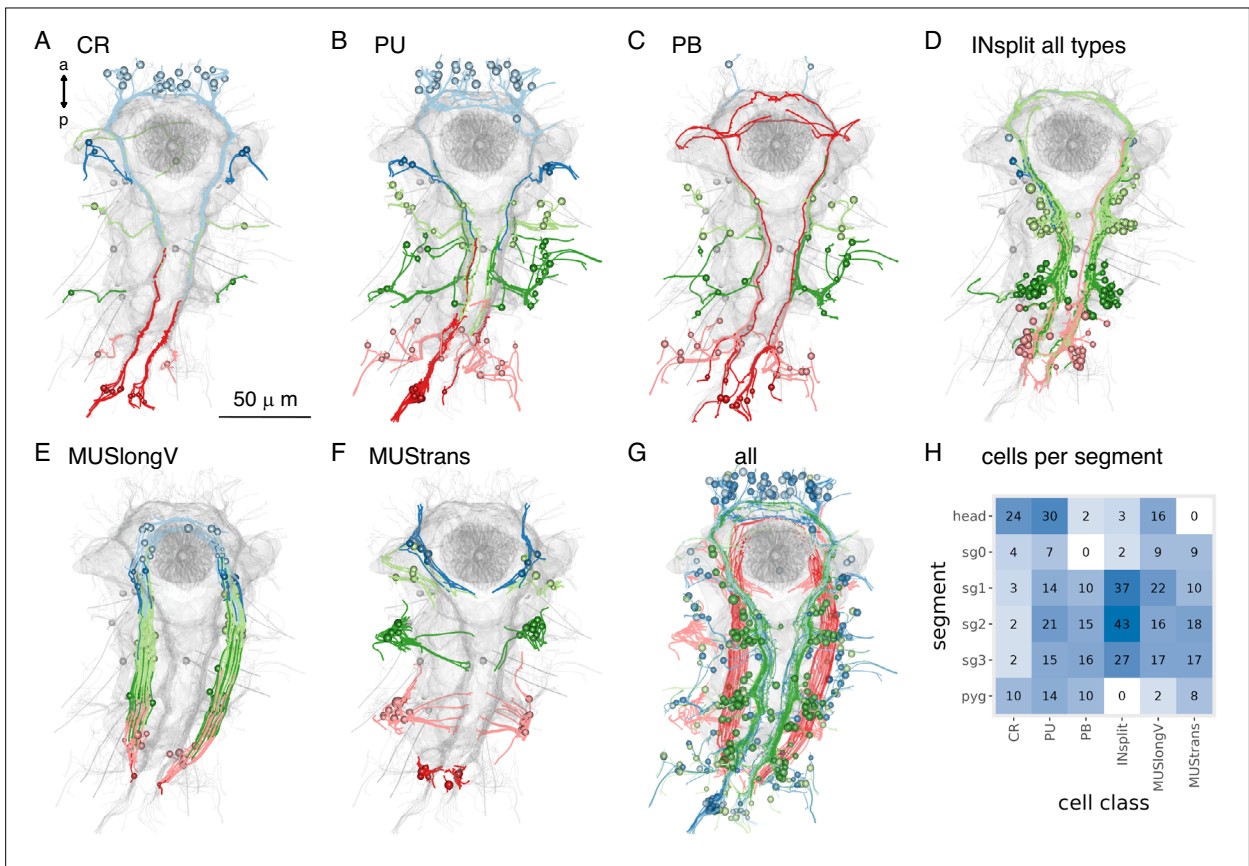

**Figure 13.** Serially homologous cell types. (**A**) Morphological rendering of all collar receptor (CR) neurons coloured by segment. (**B**) All penetrating uniciliated (PU) neurons coloured by body segment. (**C**) All penetrating biciliated (PB) neurons coloured by body segment. (**D**) All INsplit neurons coloured by body segment. (**E**) All ventral longitudinal muscles coloured by body segment. (**F**) All transverse muscles coloured by body segment. (**G**) All segmentally iterated cell classes in ventral view. (**H**) Number of cells per body segment for the six segmentally iterated cell types or cell-type families. In (**A–G**) the yolk outline, the stomodeum, the aciculae, and the neuropil of the mechanosensory girdle are shown in grey for reference.

Among the direct targets of chaeMech neurons, the INsplitCR interneurons receive the largest number of chaeMech synapses (*Figure 15E*). These neurons are also postsynaptic to the collar receptors (CR) that mediate a hydrodynamic startle response characterised by parapodial elevation and the extension of the chaetae (*Bezares-Calderón et al., 2018*). At the same time, chaeMech do not or only weakly synapse on INrope and INCM neurons, which are one of the main targets of CRs and INchaeMech neurons lack synaptic inputs from CRs. This shows that the postsynaptic circuits of CRs and chaeMechs are only partially overlapping. The shared innervation of INsplitCR by CRs and chaeMechs suggests that there may be proprioceptive inhibitory feedback during the startle response provided by chaeMechs through the INsplitCR cells.

### Parallel and converging mechanosensory circuits

The strongest postsynaptic partners of the diverse girdle mechanosensory neurons are ipsilateral interneurons with a bifurcating projection (presudounipolar morphology). This family of interneurons — collectively referred to as INsplit — could be subdivided into several distinct cell types (INsplitCR, INsplitPB, INsplitPBant, INsplitPB-RF/Ya, INsplitPUh, INsplitBronto, INCM)(*Figure 16A–H*; *Bezares-Calderón et al., 2018*).

INsplit neurons occur in all four trunk segments and in the head (INsplitPUh) and the distinct types have unique synaptic connectivity. PB and interparaPM neurons specifically target INsplitPB and represent their main input (*Figure 16B and E*). SNbronto and chaeMech neurons both synapse on the INsplitBronto interneurons (*Figures 15E, F and 16C*), which have no other major presynaptic partners. CR and chaeMech neurons synapse on INsplitCR (*Figure 16A*) while CR neurons also target INCM (*Bezares-Calderón et al., 2018*). Some head PU neurons synapse on INsplitPUh (*Figure 16D*).

Among trunk neuron types, INsplitCR and INsplitPB cells are highly ranked based on the number of pre- and postsynaptic partners, suggesting an integrating function (*Figure 12—figure supplement 2*).

INsplit neurons are premotor neurons, with connections to several trunk motoneuron types that in turn innervate glands, ciliated cells, muscles, and pigment cells (*Figure 16I*, *Figure 16—figure supplement 1*). Mechanosensory neurons thus provide input to all effector systems in the body.

## Discussion

### A whole-body resource for *Platynereis*

Here, we described a whole-body connectome for the segmented 3-day-old larva of the marine annelid *Platynereis dumerilii*. *Platynereis* is the third species for which such a whole-body resource is available, after *C. elegans* and *C. intestinalis* (*Cook et al., 2019*; *Ryan et al., 2016*; *White et al., 1986*). The power of such whole-body approaches lies in their comprehensive nature encompassing not only the nervous system but also all effectors and other cells. Overall, we identified, annotated, and spatially mapped 294 cell types and a total of 9162 cells.

*Platynereis* is distinguished from the nematode and tunicate larval connectomes by the complexity of its nervous and effector systems. The nervous system contains an order of magnitude more neurons. The effector system is multi-modal and includes muscles, locomotor multiciliated cells, glands, and pigment cells. The musculature is composed of 53 distinct muscle cell types, is segmental, and extends to the parapodia that are supported by a chitin-based endoskeleton (aciculae) (*Jasek et al., 2022*).

### Organisation of the connectome

The overall organisation of the nervous system is feed-forward, with information flow from a large diversity of sensors through interneurons to effectors. Most sensory-motor connections are relatively shallow, including direct sensory-motor neurons. However, we identified many areas of recurrent connectivity suggesting internal processing beyond sensory-motor arcs. One example is the ciliomotor system that is driven by a rhythmic pacemaker circuit (*Verasztó et al., 2017*). The endogenous activity generated by this circuit is modified by sensory inputs such as hydrostatic pressure or UV light (*Calderón et al., 2023*; *Jokura et al., 2023*).

We also identified several potential sites of multisensory integration suggesting extensive cross-talk between distinct channels of sensory processing. One experimentally characterised example is a cross-talk between phototaxis and UV avoidance (*Verasztó et al., 2018*). We have now found that the

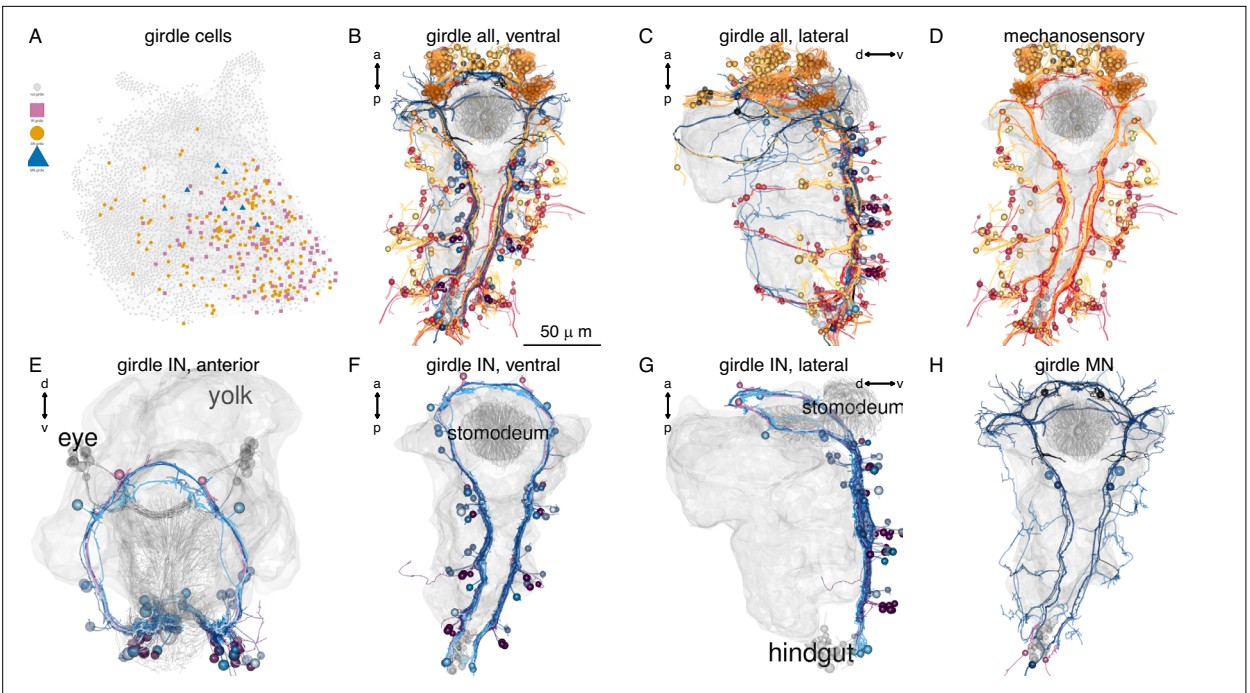

**Figure 14.** The mechanosensory girdle. (**A**) The connectome graph with cells of the mechanosensory girdle highlighted. (**B, C**) All cells of the mechanosensory girdle in ventral (**B**) and lateral (**C**) views. (**D**) Sensory neurons of the mechanosensory girdle. (**E, F**) Interneurons of the mechanosensory girdle in anterior (**E**), ventral (**F**), and lateral (**G**) views. (**H**) Motoneurons of the mechanosensory girdle. In (**B–H**), the yolk outline, the stomodeum, and the hindgut cells are shown in grey for reference. In (**E**), the visual eye photoreceptors are also shown in grey.

The online version of this article includes the following figure supplement(s) for figure 14:

**Figure supplement 1.** All cell types within the mechanosensory girdle.

**Figure supplement 2.** Cross-section of the ventral nerve cord.

eyespot R1 and adult eye photoreceptors converge on the INR interneurons (*Figure 7G and H*). This suggests a further layer of integration of light-sensory pathways. The connectome now enables the systematic identification of similar circuit motifs for sensory cross-talk with predictive value.

The connectome also shows a modular organisation and can be subdivided into functionally distinct sub-networks. These sub-networks are strongly connected within themselves, but there are also extensive connections between the modules. Such inter-module connections may establish a hierarchy of behaviours, e.g., by cross-inhibition.

The mushroom bodies also show a parallel feed-forward organisation with minimal feedback. This could be due to the developmental snapshot we acquired for the 3-day-old stage. Mushroom bodies grow in size and form a morphologically clearly recognisable region only in later-stage larvae and juveniles (*Tomer et al., 2010*). This growth is likely accompanied by circuit maturation. However, we think that the mushroom body cell types and circuits we reconstructed already show a functional circuit architecture. The MB cell types and their connections show left-right stereotypy and specificity of connectivity. It may be that annelid mushroom bodies can support associative learning by this multilayer perceptron-like organisation. In the cephalopods *Sepia officinalis* and *Octopus vulgaris*, similar feedforward information flow characterises the vertical lobe, a learning centre (*Shomrat et al., 2011*). The circuitry here has a simple fan-out fan-in architecture that in *Octopus vulgaris* shows further interconnections between the parallel feedforward networks (*Bidel et al., 2023*). In *Platynereis*, we only identified very weak connections between the parallel networks at the level of the MB projection neurons.

## Functional predictions

The connectome also enables the generation of specific circuit-level hypotheses about neuronal control and integration in the *Platynereis* larva.

The analysis of the mechanosensory systems, for example, led to several new functional predictions. The convergence of mechanosensory pathways to the MNspinning motoneurons of the exocrine spin-Glands indicates that glandular secretion is regulated by mechanosensory inputs to the parapodia. These circuits could regulate tube secretion by the benthic juvenile worms.

We also identified a potential site of mechanosensory feedback that may regulate the startle response. During a vibration-induced startle, the parapodia and their chaetae are extended, potentially activating the chaeMech chaetal mechanoreceptors. The chaeMech neurons synapse on the INsplitCR cells that are also a main target of the vibration-sensing CR neurons. An inhibitory signal from chaeMech to INsplitCR could lead to the controlled cessation of a startle.

By comprehensive tracing and cell annotation, we also identified a unique class of pigment-motor neurons (MC3cover, cioMNcover, pygPBunp). The direct synaptic innervation of pigment cells suggests that pigmentation may be under neuronal control in *Platynereis* larvae. We do not know how neuronal inputs may influence pigment cells. One possibility is that synaptic signals induce the movement or bleaching of the pigment vacuoles. In the cover cells, we occasionally observed the rapid disappearance of the pigment granules. The fact that at least one of the pigment-motor cells (pygPBunp) is also directly mechanosensory (*Bezares-Calderón et al., 2018*) suggests that mechanical cues may alter pigmentation.

Network analysis also identified various hub neurons of potential functional importance. For example, in the brain, INRGWa and INW interneurons integrate a large number of inputs. Likewise, in the trunk mechanosensory system, INsplitCR and INsplitPB neurons receive many distinct inputs. Multi-pathway convergence can occur both at the level of interneurons and on motoneurons (e.g. MNant, vMN, MNspinning).

## Circuits for whole-body coordination

The *Platynereis* whole-body connectome resource highlights the value of having access to comprehensive circuit and cell-type information. In the *Platynereis* larva, almost every functional module spans several body regions and often the entire body.

The coordinated closure and beating of cilia along the body segments is ensured by giant ciliomotor neurons that innervate target cells in multiple segments (*Verasztó et al., 2017*). The secretion of spinGlands across segments is likely also coordinated since the two MNspinning neurons innervate the four spinning glands in segments 2 and 3. Among the muscle motoneurons, the vMNs MN1 and MN2 innervate ventral and dorsal longitudinal muscles (MUSlong) in each segment. MNspider cells synapse on contralateral muscles in two consecutive segments and the muscle targets of MNcrab and MNbow neurons span three segments (*Bezares-Calderón et al., 2018*). In the pigment-motor system, the cioMNcover cells each innervate half the ring of the cover cells and pygPBunp innervates the entire ring, also suggesting coordinated regulation.

Intersegmental coordination is also apparent at the level of sensory and interneurons. For example, individual chaeMech cells can synapse on interneurons in three segments. INrope and INchaeMech interneurons target motoneurons across three segments, INsplitCRATO, INsplitBronto, and INsplitVent in two segments. The globally reaching INsplitPBant cells have postsynaptic partners in all four segments and the pygidium.

These examples demonstrate the importance of a whole-body approach in connectomics to understanding circuit function and behaviour. Partial connectomes could not deliver satisfactory circuit explanations for any of these systems. We do not expect this to be different for other animals.

## Circuit evolution by duplication and divergence

An exciting perspective in connectomics is to learn about circuit evolution by the analysis of comprehensive datasets that integrate anatomy and connectivity. Our reconstructions identified a potential case for circuit evolution by duplication and divergence (*Roberts et al., 2022*; *Tosches, 2017*), a concept that originated from genetics (*Ohno, 1970*).

The mechanosensory girdle contains several morphologically similar mechanosensory neurons bearing penetrating cilia at the tip of their sensory dendrites (PB, CR, interparaPM, PU, and pygPBunp). These cells likely represent evolutionarily related sister cell types (*Arendt, 2008*). Some of these neurons also show overlapping gene regulatory signatures — supporting their classification as

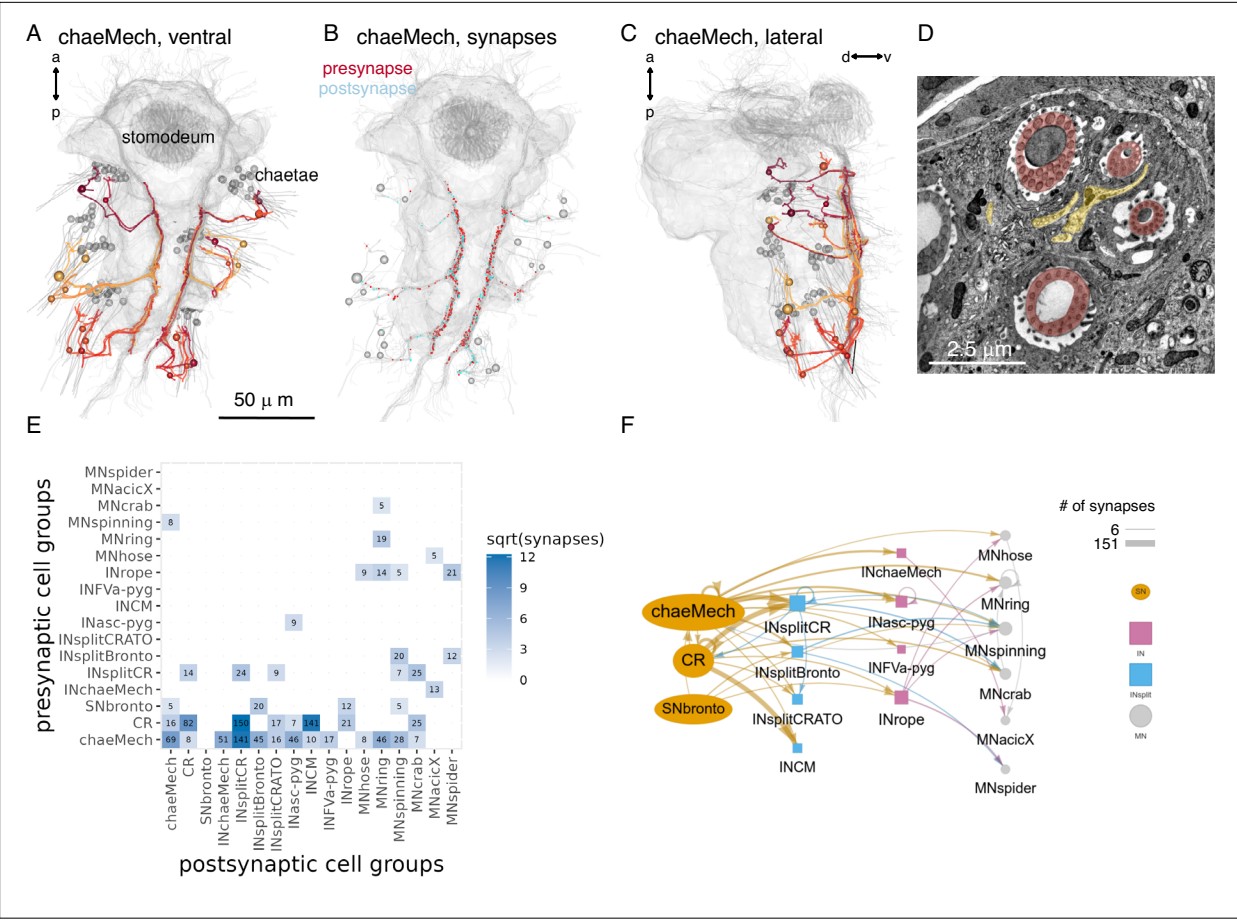

**Figure 15.** Chaetal mechanosensors and their circuits. (**A**) Morphological rendering of chaeMech neurons, ventral view. (**B**) Presynaptic (red) and postsynaptic (cyan) sites of chaeMech neurons, ventral view. The soma of the chaeMech cells is shown in grey. (**C**) Lateral view of chaeMech neurons. (**D**) Transmission electron microscopy (TEM) image of the sensory dendrites of chaeMech neurons (yellow highlight) surrounding the chaetae (red highlight). (**E**) Grouped synaptic connectivity matrix of chaeMech, CR, and SNbronto neurons and their downstream targets. (**F**) Same connectivity information as in (**E**) represented as a network. In (**A–C**), the yolk outline, stomodeum, and the mechanosensory girdle neuropil are shown in gray for reference. In (**A**, **C**), the aciculae and chaetae are also shown. *Figure 15—source data 1*. Connectivity matrix for the network in panels **E**, **F**.

The online version of this article includes the following source data for figure 15:

**Source data 1.** Connectivity matrix for the network in panels E, F with the grouped synaptic connectivity matrix of chaeMech, CR and SNbronto neurons and their downstream targets.

sister cell types — as they all can be labelled by the same *PKD2-1* transgene (PB, CR and pygPBunp) (***Bezares-Calderón et al., 2018***).

These mechanosensory neurons synapse on distinct subsets of INsplit types. Based on their morphological similarity, INsplit cells are also likely related among themselves.

This pattern suggests that these parallel systems may have evolved through the process of circuit duplication and divergence. According to this model, the diversification of mechanoreceptors may have been paralleled by the diversification of their postsynaptic INsplit neurons. This enabled sensory refinement in parallel with the evolution of matched downstream circuits and behaviours.

The duplication and divergence of cell-type sets also characterised the evolution of the vertebrate cerebellum (***Kebschull et al., 2020***). Duplication and divergence of circuits was also proposed as a possible mechanism for the evolution of brain pathways for vocal learning in song-learning birds, spoken language in humans (***Chakraborty and Jarvis, 2015***), and other circuits (***Roberts et al., 2022***). Further support for this model in *Platynereis* could come from comprehensive gene expression analyses for the distinct mechanosensory and INsplit neurons.

# Connectomics informs the evolution of the annelid segmental body plan

Our dataset represents, to our knowledge, the first whole-body connectome of a segmented animal. These data also inform our understanding of the evolution of the annelid body segments and nervous system, both long-standing questions in evolution and development (*Balfour, 1881*; *Nielsen et al., 2018*; *Nielsen, 2005*; *Sedgwick, 1884*; *Starunov et al., 2015*; *Steinmetz et al., 2011*).

By mapping cell-type distributions across segments, we found evidence for segmental homology overlain by a heteromeric pattern with segment-specific cell types. The serial homology of the cryptic segment (sg0) with the other three trunk segments (sg1-3) (*Saudemont et al., 2008*; *Steinmetz et al., 2011*) is confirmed by neuron and muscle types shared between these four segments. In addition, the pygidium and the head may have metameric origin. The pygidium has a coelomic cavity and muscle cell types (MUSlong, MUStrans) shared with the other trunk segments (*Jasek et al., 2022*; *Starunov et al., 2015*). The segmental origin of the annelid head is suggested by a Cambrian annelid fossil where the head is formed by a segment with parapodia and chaetae (*Parry et al., 2015*).

The metameric origin of all six body regions receives additional support from our connectome reconstructions. We identified muscle cell types (MUSlongV, MUStrans), sensory neuron types (CR, PB, PU), and one family of interneurons (INsplit) shared by five or six of the body regions. One interpretation of this pattern is that the annelid body evolved from an organisation with six homomeric regions that subsequently diversified but retained a core set of homologous cells. Further tests of this model could come from future whole-body connectomes of other annelid species.

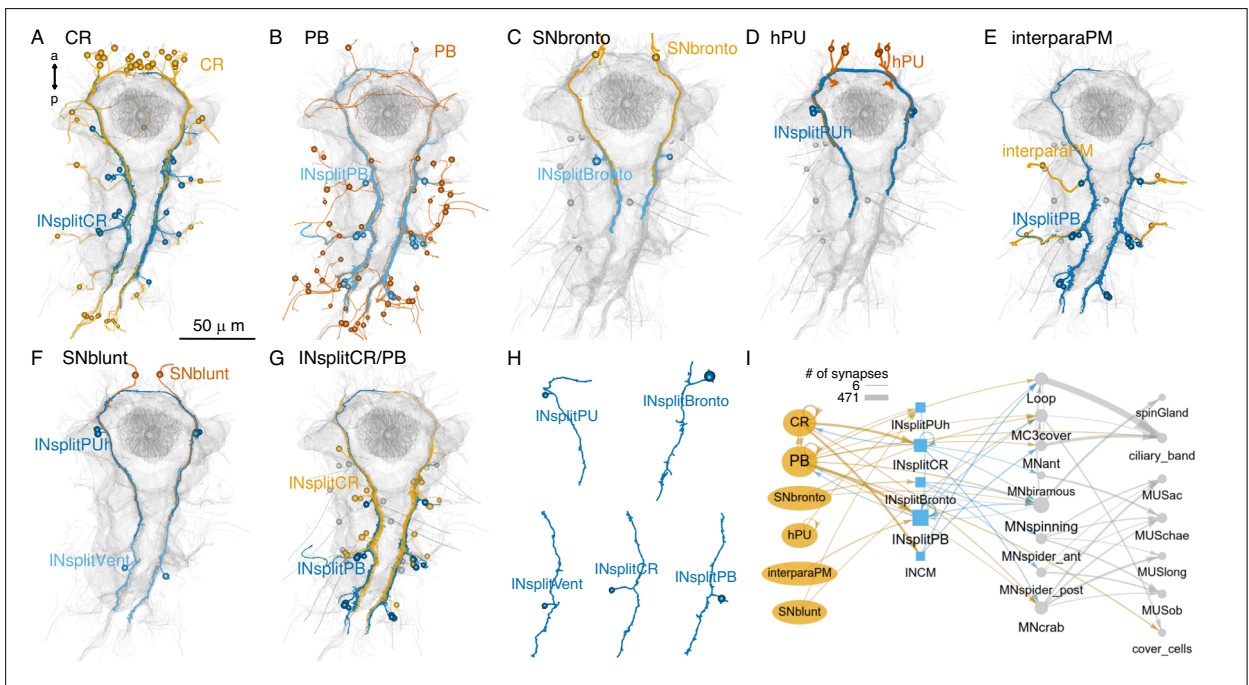

**Figure 16.** Parallel systems of mechanosensory neurons and their INsplit targets. (**A**) Morphological rendering of all collar receptor (CR) neurons and their INsplitCR targets. (**B**) All penetrating biciliated (PB) neurons and their INsplitPB targets. (**C**) SNbronto neurons and their INsplitBronto targets. (**D**) Head-penetrating uniciliated (hPU) neurons and their INsplitPUh targets. (**E**) Interparapodial penetrating multiciliary (inerparaPM) neurons and their INsplitPB targets. (**F**) SNblunt neurons and their INsplitPUh and INsplitVent targets. (**G**) INsplitCR and INsplitPB neurons. (**H**) Example morphologies of pseudounipolar INsplit neurons. Spheres represent the position of cell somas. (**I**) Grouped connectivity diagram of the six mechanosensory cell classes, their direct INsplit targets, and downstream motoneurons and effectors. Only connections with >5 synapses are shown. In (**A–G**), the stomodeum, the aciculae, and the yolk outline are shown for reference. *Figure 16—source data 1*. Connectivity matrix for the network in panel **I**.

The online version of this article includes the following source data and figure supplement(s) for figure 16:

**Source data 1.** Connectivity matrix for the network in panel I with the grouped connectivity data of the six mechanosensory cell classes, their direct INsplit targets and downstream motoneurons and effectors.

**Figure supplement 1.** Connectivity of mechanosensory neurons.

The identification of a mechanosensory girdle in *Platynereis* reminded us of another classic hypothesis about bilaterian body-plan evolution.

In 1881, Francis Balfour put forward the hypothesis that the ancestral nervous system from which the nervous system of arthropods, molluscs, and other invertebrates derived was 'a circumoral ring, like that of Medusae, with which radially-arranged sense-organs may have been connected.' He posits a transition scenario in which a 'circumoral nerve-ring, if longitudinally extended, might give rise to a pair of nerve-cords united in front and behind' (*Balfour, 1881*).

In 1884, Sedgwick further developed this model and derived the bilaterian mouth and anus through the fusion of the lateral lips of the gastric slit of a radially symmetric animal (*Sedgwick, 1884*). Metameric segmentation in this model evolved from the mesenteries and somites from gut pouches of a radial ancestor. For a thorough modern treatment of this Balfour-Sedgwick model — also referred to as the amphistomy theory — see *Nielsen et al., 2018*.

Our connectome reconstructions in the *Platynereis* larva are compatible with the Balfour-Sedgwick theory. The mechanosensory girdle looping around all six homologous body segments can be interpreted as a derivative of a circumoral nerve ring looping around homomeric body regions. The girdle could correspond to radially arranged mechanosensory organs around the cnidarian oral opening (e.g. *Singla, 1975*; *Figure 17*). Alternative transition scenarios, such as the evolution of a new opening at the aboral pole of a radial ancestor and de novo evolution of a mechanosensory, circumoral nerve ring, seem less plausible.

Testing this model would require detailed reconstructions of cnidarian circumoral nervous systems. We predict that there may be bifurcating interneurons in cnidarians that are postsynaptic to circumoral mechanosensors and project in two directions along the oral opening (similar to INsplit neurons).

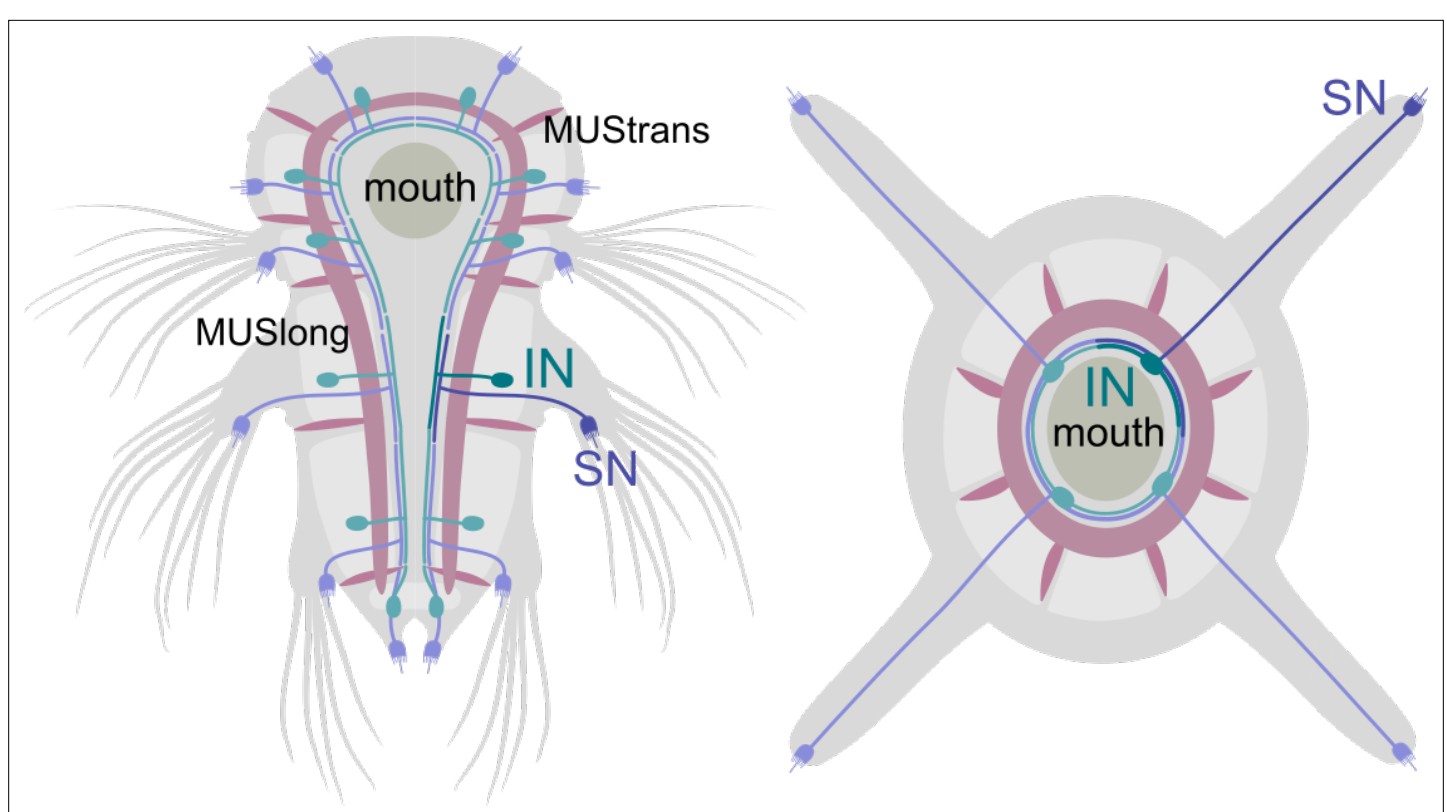

**Figure 17.** Origin of the mechanosensory girdle by amphistomy. Hypothetical homology of the circumoral nervous system in a radially symmetrical ancestor of bilaterians and the mechanosensory girdle.

# Materials and methods

## Specimen preparation, transmission electron microscopy, and image processing

Fixation and embedding were carried out on a 72 hpf *Platynereis* larva (HT9-4) as described previously (*Conzelmann et al., 2013*). Serial sectioning and transmission electron microscopy were done as described in *Shahidi et al., 2015*. The section statistics for the HT9-4 (NAOMI) specimen were previously described (*Randel et al., 2015*). The serial sections were imaged on a FEI TECNAI Spirit transmission electron microscope with an UltraScan 4000 4X4k digital camera using Digital Micrograph acquisition software (Gatan Software Team Inc, Pleasanton) and SerialEM (*Schorb et al., 2019*). The images for the HT9-4 projects were scanned at various pixel resolutions: 5.7 nm/pixel, 3.7 nm/pixel, and 2.2 nm/pixel. Images were stitched and aligned with TrakEM2 (*Cardona et al., 2012*).

We used the collaborative annotation toolkit CATMAID for tracing, annotation, and reviewing of skeleton annotations (*Saalfeld et al., 2009*; *Schneider-Mizell et al., 2016*).

Due to contrast and focus problems in the main dataset, we had to re-image certain layers at higher resolutions, to allow tracing of neurons. These re-imaged series were made into independent CATMAID projects. This included five extra projects, taken at various points throughout the main dataset. All layers, including the information on lost layers, immunogold labelling and re-imaging are listed in *Supplementary file 2*. The largest of these, Plexus_HT-4_Naomi_project__372–4013, consisted of 1407 layers at a resolution of 2.2 nm. This stack mostly focused on the brain plexus and the ventral nerve cord where most neurites and synapses occur. Other projects consisted of three jump/gap regions that required not only high resolution but also better realignment. One set contained all the immunogold-labelled layers (*Shahidi et al., 2015*) that were not included in the main aligned dataset. These layers had very low contrast due to the immunolabelling procedure and, therefore, required higher resolution imaging. All projects were first created and processed in TrakEM2 and then exported as flat jpeg images into CATMAID.

## Image-stack realignment and transformation of spatial data in CATMAID

To improve the traceability of the vEM stack in all areas, we realigned the raw TEM images to improve the previously reported alignment of the volume (*Randel et al., 2015*). First, we opened the original project with the TrakEM2 plugin for FIJI (ImageJ) (version 2.0.0-rc-15/1.49k/Java 1.6.0_24 (64-bit) – 2014). We set the region of interest (ROI) to width: 25792, height: 28800, x-shift: 20885, y-shift: 11928. All 4846 layers were then exported as flat TIFF images with a scale of 100%, 8-bit grayscale, no background color (0). A new blank TrakEM2 project was then created in FIJI, and the exported TIFF images were imported using the project import function 'import sequence as grid.' The image filters were previously applied to the images, and, therefore, were not required. First, an 'Affine' alignment was applied using the following parameters: least squares (linear feature correspondences) mode, choosing entire layer range with first layer as reference, using visible images only and no propagation, initial Gaussian blur of 1.6 pixels, 3 steps per scale octave, minimum image size of 64 pixels and maximum of 2048 pixels, feature descriptor size of 8, feature descriptor orientation bins of 8, closest ratio of 0.92, with clear cache selected, feature extraction threads of 30, maximal alignment error of 100 pixels, minimal inlier ratio of 0.20, minimal number of inliers of 12, expected transformation as Affine, testing multiple hypotheses with tolerance of 5.00 pixels, testing maximal layer neighbour range of five layers, giving up after five failures, desired transformation as Affine, regularizing model, maximal iteration of 1000, maximal plateau width of 200, regularizer as Rigid, and lambda of 0.10. Next, two iterations of Elastic alignment were applied using the following parameters: block matching layer scale of 0.05, search radius of 200 pixels, block radius of 2000 pixels (increased to 2400 pixels during second iteration), resolution of 60, correlation filters with minimal PMCC r of 0.10, maximal curvature ratio of 1000, maximal second best r/best r of 0.90, using local smoothness filter, with approximate local transformation as Affine, local region sigma of 1000 pixels, absolute maximal local displacement of 10 pixels, relative maximal local displacement of 3.00, as pre-aligned layers, testing maximal of 4 layers, approximate transformation as Rigid, maximal iterations of 1000, maximal plateau width of 200, spring mesh stiffness of 0.01, maximal stretch of 2000 pixels, maximal iterations of 3000, maximal plateau width of 200, using legacy optimizer. After each alignment procedure, the project

was saved as an XML file with a different name. Finally, images were exported for CATMAID using TrakEM2 in FIJI (version 2.0.0-rc-69/1.52 p/Java 1.8.0_172 (64-bit) – 2019). After the realignment, the transformation of all traced neuron skeletons in CATMAID was necessary to match the newly applied image alignment transforms. To achieve this, we used a logic script (Tom Kazimiers, Kazmos GmbH) that applied a realignment transformation process from the TrakEM2 XML project file to the CATMAID data. This script was added as a management command in CATMAID (https://github.com/catmaid/CATMAID/blob/master/django/applications/catmaid/management/commands/catmaid_update_tracing_data_using_trakem2_xml.py). When applied, these transformation offsets were within ± 1 pixel accuracy for all parented nodes. The code is available in the CATMAID GitHub repository, commit e25debb (https://github.com/catmaid/CATMAID/commit/e25debb, RRID:SCR_006278).

## Neuron tracing, synapse annotation, and reviewing

To digitally reconstruct every neuron in the serial TEM dataset of the three-day-old larva, we used the collaborative web application CATMAID (*Saalfeld et al., 2009*; *Schneider-Mizell et al., 2016*) installed on a local server.

The total construction time of all skeletons was over 2970 hr with an additional 782 hr of review time.

To mark the position of cell somas, we tagged the centre of each nucleus in the volume. At the approximate centre of each nucleus, we changed the radius of a single node according to the soma size in that layer. All skeletons were rooted on the soma and the node was tagged with 'soma'.

Ultrastructural features (number and orientation of microtubules, electron density of the cytoplasm and vesicles, ER structure) and a high-resolution dataset of the neuropil and the ventral nerve cord aided tracing through low-quality layers and gaps.

We identified synapses based on a vesicle cloud close to the plasma membrane. Most synapses were visible in consecutive layers (for example, images see [*Randel et al., 2014*] and browse the data). We also checked for the proximity of mitochondria in the same arbour in case of ambiguous synapses — a requirement supported by quantitative connectomic data in *Drosophila* (*Schneider-Mizell et al., 2016*). The systematic review of all neurons belonging to a cell type was done by one or multiple reviewers until close to 100% was reached for every cell. Cells were further checked in the 3D widget to split implausible skeletons. Synapses were reviewed multiple times, from both the pre- and postsynaptic arbours.

## Cell nomenclature and annotations

All cells have a unique name. We named neurons based on their type (e.g. sensory or motor; SN, MN) cell body position (left or right, head, or trunk segment), axonal morphology, neuropeptide expression, and other specialisations (e.g. sensory morphology). Cells of the same type have similar names, distinguished by body position indicators and numbers. These indicators follow the general cell-type name separated by the first _ symbol in the name string. We endeavoured to give names that were easy to remember. The name of many sensory neurons start with SN followed by a specific term (e.g. blunt, bronto, stiff). Interneurons often start with IN and motoneurons with MN. There are exceptions, including neurons with known function (e.g. PRC for photoreceptor cells) and neurons with prominent morphology (e.g. Loop). Segmental position (sg0-3) and body side (l or r) is indicated in the name of most neurons. Non-neuronal cells were named based on anatomical terms (e.g. prototroch) or by abbreviations (e.g. EC for epithelial cell). We have updated the numbering of the prototroch cells (by counter-clockwise rotation; prototroch_1 to prototroch_2 etc.) to better match the developmental cell lineage and recent literature (*Poon et al., 2025*; *Vopalensky et al., 2019*). The old names we used in previous publications (*Randel et al., 2015*; *Verasztó et al., 2017*) were kept as annotations in the format 'old_name:name:DOI'.

All cells have multiple annotations, which can be used to query the database in CATMAID or through the CATMAID API (e.g. in R b the catmaid package). Neurons belonging to a cell-type category were annotated with the generalist annotation 'celltype' and a cell-type-specific annotation (e.g. celltype23 for the INpreMN neurons). Non-neuronal cells belonging to a cell-type category were annotated with the generalist annotation 'celltype_non_neuronal' and a cell-type-specific annotation (e.g. celltype_non_neuronal23 for the acicula cells). Neurons were also annotated with descriptors of their projection morphologies (e.g. commissural, ipsilateral, pseudounipolar etc.), neuron class (Sensory neuron,

sensory-motor neuron, sensory-neurosecretory neuron, interneuron, inter-motorneuron and motor-neuron [note the 'r,' which we kept for backward compatibility]). In CATMAID, we recommend the use of regular expressions for searching annotations e.g., ^motorneuron$ to retrieve exact matches. Differentiating neurons with immature sensory dendrites or axonal projections with axonal growth cones and with no or few synapses were annotated 'immature neuron' (402 cells). Ascending trunk neurons and descending head neurons traversing the circumesophageal connectives were annotated 'head-trunk.' Neurons with a soma in the head and a descending decussating axon were annotated with 'decussating.' Cells were also annotated according to the location of their soma in a certain body region (head — as 'episphere,' trunk — as 'torso,' 'pygidium'), body side ('left_side,' 'right_side'), segment ('segment_0,' etc.), and germ layer (ecto-, meso-, endoderm).

### Criteria for including cells in the connectome

When defining the connectome, we aimed at including differentiated cells and skeletons connected with at least three synapses to the main graph. We used the script 'connectome_from_CATMAID.R' to derive the final full connectome graph. First, we fetched all synaptic connectors and their pre- and postsynaptic partners. Each single synapse was assigned a weight of one and edges connecting the same nodes in the same direction were summed. We then removed all vertices from the graph with <3 synapses. We checked for connected components and selected the largest subgraph.

### Quantitative analysis of neuron morphologies

For quantitative neuroanatomy, we used functions of CATMAID as implemented in the catmaid (catmaid-package {catmaid}: R access to the API for the CATMAID web image annotation tool) and nat (nat-package {nat}: Analyse 3D biological image data especially neurons) packages. For morphological rendering and quantitative analysis, skeletons were first smoothed with the smooth_neuron function (smooth_neuron {nat}: Smooth the 3D coordinates of a neuron skeleton) with a sigma of 6000 nm. Neuronal cable length was calculated on trimmed skeletons. Each twig shorter than 2 mm was removed with the prune_twigs {nat} function. Cable length then gives the length of the trimmed skeletons in microns (Figure1_fig_suppl2.R code in the code repository). For Sholl analysis, we used the sholl_analysis function (sholl_analysis {nat}: Perform a Sholl analysis on neuron skeletons) on untrimmed, smoothed skeletons. The radial density of input and output synapses was calculated in the CATMAID web interface and the data were saved as .csv and plotted in R (Figure1.R code).

### Network layout

The layout of the full connectome was generated by force-field-based clustering. We used the Force Atlas tool in Gephi 0.10.1 (*Bastian et al., 2009*). The inertia was set to 0.1, repulsion strength was 35, attraction strength was 10, maximum displacement was 5, gravity was 20, speed was 5, and the attraction distribution option was on. The 'auto-stabilise function' was off. Towards the end of the clustering the 'adjust by sizes' option was also selected. To prevent node overlap, we then ran the 'Noverlap' function. Node positions from this Gephi layout were imported and further graph manipulations (including node colouring) were carried out in R.

### Network analysis

We used CATMAID (several releases), Gephi 0.10.1, and R for network analysis. For graph analysis and visualisation, we used R and the iGraph, tidygraph, visNetwork, and networkD3 packages (*Allaire et al., 2017*; *Almende et al., 2019*; *Csardi and Nepusz, 2006*).

### Data analysis and plotting

For data analysis, we used R and RStudio (various releases) (*Posit team, 2023*). For data handling and plotting, we endeavoured to adhere to the practices and packages of the Tidyverse (*Wickham et al., 2019*). Data plotting was done with the ggplot2 package (*Wickham et al., 2016*). See also the 'versionInfo.txt' and 'sessionInfo.txt' files in the repository for full package and version information.

## Acknowledgements

This research was funded by a Wellcome Trust Investigator Award 214337/Z/18/Z. The research has also been supported by a grant from the Deutsche Forschungsgemeinschaft (JE 777/3—1). This project

has received funding from the European Research Council (ERC) under the European Union's Horizon 2020 research and innovation programme (grant agreement No 101020792). We thank Nobuo Ueda, Nadine Randel, James David Beard, and Sara Mendes for contributing to tracing and Konrad Heinz for helping to implement an R workflow with the Natverse package. We thank Tom Kazimiers for support with the realignment of the stack.

## Additional information

### Competing interests

Gáspár Jékely: Reviewing editor, eLife. The other authors declare that no competing interests exist.

### Funding

| Funder | Grant reference number | Author |
|---|---|---|
| Wellcome Trust | 10.35802/214337 | Gáspár Jékely |
| Deutsche Forschungsgemeinschaft | JE 777/3-1 | Gáspár Jékely |
| European Research Council | 10.3030/101020792 | Gáspár Jékely |

The funders had no role in study design, data collection and interpretation, or the decision to submit the work for publication. For the purpose of Open Access, the authors have applied a CC BY public copyright license to any Author Accepted Manuscript version arising from this submission.

### Author contributions

Csaba Verasztó, Data curation, Formal analysis, Validation, Investigation; Sanja Jasek, Data curation, Software, Formal analysis, Validation, Investigation, Visualization; Martin Gühmann, Luis Alberto Bezares-Calderón, Elizabeth A Williams, Data curation; Réza Shahidi, Resources, Data curation; Gáspár Jékely, Conceptualization, Data curation, Software, Formal analysis, Supervision, Funding acquisition, Validation, Visualization, Methodology, Writing – original draft, Project administration, Writing – review and editing

### Author ORCIDs

Sanja Jasek (ID) https://orcid.org/0000-0001-6844-4319
Gáspár Jékely (ID) https://orcid.org/0000-0001-8496-9836

Reviewer #1 (Public review): https://doi.org/10.7554/eLife.97964.3.sa1
Author response https://doi.org/10.7554/eLife.97964.3.sa2

## Additional files

### Supplementary files

MDAR checklist

Supplementary file 1. List of all neuronal and non-neuronal cell types, their main annotations, including morphological, cell-class, segment, ganglion or sensory organ, neurotransmitter, and neuropeptide phenotype, main synaptic partners, and reference.

Supplementary file 2. Summary of all electron microscopy (EM) layers including the information on lost layers, immunogold labelling, and re-imaging.

### Data availability

The EM image stack with all traces and annotations are available at https://catmaid.jekelylab.ex.ac.uk and can be queried in CATMAID or via the CATMAID application programming interface (API) with the catmaid/natverse packages (*Bates et al., 2020*). The dataset includes all EM images (in jpg format), skeletons, meshes, node tags, connectors (both synapses and desmosomes), and

annotations. We also provide all the R scripts we used for data retrieval and the generation of the figures (*Verasztó et al., 2025*). All plots, diagrams (including anatomical renderings), and figure layouts should be fully reproducible by using the R scripts provided. Scripts were mostly organised by figure, except some generalist scripts, e.g., to load libraries, CATMAID access data, and general functions ('Natverse_functions_and_conn.R').

The following dataset was generated:

| Author(s) | Year | Dataset title | Dataset URL | Database and Identifier |
|---|---|---|---|---|
| Verasztó C, Jasek S, Gühmann M, Bezares-Calderón LA, Williams EA, Shahidi R, Jékely G | 2024 | Catmaid database of the three day old Platynereis larval connectome | https://catmaid. jekelylab.ex.ac.uk | CATMAID, Naomi_ realigned |

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
