## [Editor Report · eLife Assessment]

This **important** study is an advancement towards the understanding of animal nervous system organization and evolution by providing an **exceptional**, high-quality and detailed description of the entire connectome of the 3-day larva of the marine annelid *Platynereis dumerilii*. It provides a wealth of data on cell type diversity and the modules that interconnect them. Its strength is the massive amount of high-quality data, although this is also partly a weakness as it can make the work difficult to read and digest scientifically. This work lays the foundations for studies on cell type diversity, segmental vs. intersegmental connectivity, and mushroom bodies, but will certainly also be of use to scientists interested in other nervous systems parts, their functions, and evolution.

---

## [Referee Report · Reviewer #1 (Public review)]

Summary:

This paper provide a resource for researchers studying the marine annelid *Platynereis dumerilii*. It is only the third whole body connectome to be assembled and thus provides a comparison with those less complex animals: the nematode *Caenorhabditis elegans* and the tunicate Ciona intestinialis. The paper catalogs all cells in the body, not just neurons, and details how sensory neurons, interneurons, motor neurons, and effector organs are connected. From this, the authors are able to extract information about the organization of different aspects of the nervous system. These include the extent of recurrent connectivity, unimodal and multimodal sensory processing, and long-range and short-range connectivity.

Several interesting conclusion are drawn, including the concept that circuit evolution might have proceeded by duplication and diversion of cell types, much as it has been posited that gene evolution has occurred. It also informs the understanding of the evolution of segmental body plans in annelids by mapping and comparing cells in each segment.

Strengths:

This paper contains a wealth of data. The raw dataset is available. The codes and scripts are provided to allow interested readers to utilize this dataset.

The analysis is painstakingly meticulous. The diagrams are organized to orient the reader to the complexities this overwhelming analysis

Weaknesses:

The strength of the paper is also its weakness. It contains so much data and analysis that it is burdensome to read and understand. There are 16 multi-panel data figures in the main text and another 38 supplemental figures and 5 videos.

The impact of the paper is diminished by its size and depth. The paper could be broken up into smaller thematic papers that would be more accessible to researchers interested in particular topics. For example, there could be a single paper on the mushroom body and another paper on the segmental organization.

Comments on revisions:

The authors have addressed all of my concerns.

---

## [Author Response]

The following is the authors’ response to the original reviews.

**Recommendations for the authors:**
Reviewing Editor Note:The two reviewers have provided thoughtful and constructive feedback that we hope will be of use to the authors to improve their manuscript.
**Reviewer #1 (Recommendations For The Authors):**
The section on "Circuit evolution by duplication and divergence" (starting on line 622) should cite:Chakraborty, Mukta, and Erich D. Jarvis. "Brain evolution by brain pathway duplication." Philosophical Transactions of the Royal Society B: Biological Sciences 370, no. 1684 (2015): 20150056.andRoberts, Ruairí JV, Sinziana Pop, and Lucia L. Prieto-Godino. "Evolution of central neural circuits: state of the art and perspectives." Nature Reviews Neuroscience 23, no. 12 (2022): 725-743.It should also reference that the concept originated from genetics:Ohno, Susumu. Evolution by gene duplication. Springer Science & Business Media, 1970

These papers have now been cited: “Duplication and divergence of circuits was also proposed as a possible mechanism for the evolution of brain pathways for vocal learning in song-learning birds, spoken language in humans [@chakraborty2015brain] and other circuits [@roberts2022evolution].”

and: Our reconstructions identified a potential case for circuit evolution by duplication and divergence [@tosches2017developmental; @roberts2022evolution], a concept that originated from genetics [@ohno1970evolution].

The terms outgoing and incoming synapses were confusing. The more common terminology is pre and postsynaptic elements. For example, in Fig 1, the label Sensory neuron outgoing and incoming was confusing because I mistakenly thought it was referring to the neurons and I could not figure out what an outgoing sensory neuron was.

We have now changed ‘incoming’ to ‘postsynaptic’ and ‘outgoing’ to ‘presynaptic’.

In L-O, there should be an indicator on the figures that they refer to the locations of synaptic sites, as it does in F.

We have now replaced the labels ‘incoming’ and ‘outgoing’ with ‘presyn’ and ‘postsyn’ for Figure 1 panels L-O to make it clear that these are synaptic sites.

Figure 2. - last panel of muscle motor - it would be helpful to have names of muscles instead of just having 5 'muscle motor' of different colors

Each muscle-motor module contains a large number and type of muscles and motor neurons. Labelling them by the name of individual muscle types is therefore not practical at this resolution. The three-day-old *Platynereis* larvae has 53 different muscle cell types. Their anatomy and classification, together with the details of motoneuron innervation have been described in detail elsewhere (Jasek et al 2022 https://doi.org/10.7554/eLife.71231).

Figure 3. D and E are hard to understand from the figure; The shading is the number of neurons; that scale should be shown somewhere.

We are not sure we understand the comment. These plots are histograms that show the distribution of the number of cells across categories. The y axis is the number of neuronal or non-neuronal cell types in each bin.

PageRank is an algorithm that Google uses. In Figure 4, it seems to be used to indicate centrality. A brief explanation in the text would be useful.

We have now added an explanation of the centrality measures used. “PageRank is an algorithm used by Google to rank webpages and scores the number and quality of the incoming links of a node [@page1999pagerank], betweenness centrality measures the number of shortest paths that pass through a node in a graph [@freeman1977set], and authority measures the extent of inputs to a node by hubs in a network [@kleinberg1999authoritative].”

Figure 5. The labels on some images are not clear. They are on top of each other and elements of the figure

We have now moved the position of the labels to minimise overlap. We have also added an interactive html file with the network shown in Figure 5 panel A to help the exploration of the network. Added: “Figure 5—source data 1. Interactive html file with the network shown in panel A.”

There are differences in line thickness in several figures, such as Figure 9 (A and B) and Figure 12 (D and I and N) that presumably means numbers of synaptic contacts. It would be useful to know what the scale is.

We have now added labels of line thickness to the networks in Figure 4, Figure 5 – figure supplement 2, Figure 9, Figure 12, Figure 7 – figure supplement 1, Figure 15 and Figure 16.

**Reviewer #2 (Recommendations For The Authors):**
(1) Suggestions for improved or additional experiments, data, or analyses.(2) Recommendations for improving the writing and presentation.Perhaps we require a comprehensive inventory detailing all the innovations compared to previous, more limited publications, particularly in relation to the 2017 publication and 2020 preprint.

We have provided this detail in Supplementary table 1 that lists all cell types. We included the reference for previously published cell types in the ‘reference’ column except for those that were also described in the 2020 preprint. The current manuscript is a greatly revised and extended version of the original 2020 preprint. In addition, in the online connectome database (https://catmaid.jekelylab.ex.ac.uk), all cell types that were previously published are annotated with the notation ‘FirstAuthor_et_al_year’.

It is a bit frustrating given the huge amount of graphs, analyses, tables, and networks that are presented in the manuscript, we do not see much of the original EM pictures except for a few examples of cell type blow-ups. It would be useful for future workers in the field to have eventually a sort of compendium of how the authors actually recognized each cell type, without having to connect to the original CATMAID annotation.

Most neuronal cell types (with the exception of some characteristic sensory neurons such as photoreceptor cells and mechanosensory cells) were not classified based on ultrastructural features, but on features of neurite morphology, body position and synaptic connectivity. It would be therefore not possible to represent most of the cell types with a single layer of an original EM picture. However, in order to make the morphological skeleton characteristics more accessible to the reader, we have now added a comprehensive website (https://jekelylab.github.io/Platynereis_connectome/) including all cell types together with their interactive 3D rendering.

“Interactive 3D morphological renderings of each cell type together with their main annotations can also be explored on a webpage (https://jekelylab.github.io/Platynereis_connectome/celltype_compendium/index.html).”

The Platynereis 3-day larva is obviously only one transient stage in the developmental cycle of the animal, and it is a very specialized stage (called metatrochophore in annelid jargon), during which the animal does not yet feed, relying instead on its copious yolk. Moreover, it is a stage whose purpose is limited to dispersion, with no complex behavior or social interaction that later stages are going to display. While this work represents a substantial leap forward in understanding neural integration in a whole animal, it must be kept in mind that compared to an adult or growing juvenile, there are likely a considerable number of cells, cell types, and neural modules missing in this larva. This is clearly not a weakness of this study per se, but readers may find it interesting to be presented with this perspective and therefore more biological details about the Platynereis life cycle and associated behaviors.Obviously, understanding how the constantly developing nervous system of a worm-like Platynereis gets reshuffled in time will be a great subject to investigate. The authors mention that the 3-day larva displays more than 4000 neuronal cells not yet differentiated. Readers may be interested in their location. Are there niches of neural stem cells? A description of what may be missing from the larva in terms of cell types compared to the adult may be useful.

We have now added further explanation into the Introduction about the early nectochaete larval stage: “The early nectochaete larva represents a transient dispersing stage in the life cycle of Platynereis. During this stage the larvae do not feed yet but rely on maternally provided yolk. Compared to the juvenile and adult stages it is expected that a considerable number of cell types will be only developing or completely missing at this stage. Three-day-old larvae do not yet have sensory palps and other sensory appendages (cirri), they do not crawl or feed and lack visceral muscles and an enteric nervous system.”

The location of developing neurons is shown in Figure 3—figure supplement 1 panel I.

Juvenile or adult cell types have not yet been described in any detail that is close to the level of detail we now provide for the nectochaete larva, therefore a meaningful comparison of cell-type complements across stages is not yet feasible.

(3) Minor corrections to the text and figures.Figure 1: "outgoing" not "outgoung" in panels M, O, Q.

Corrected

Line 128: We may need a precise definition of "cable length".

We have included a definition of cable length in the Methods section under a new subheading ‘Quantitative analysis of neuron morphologies’.

In all Figures: information on the orientation of the worm's view is sometimes missing in figures, which could make interpretation difficult for the reader, especially for anterior views with no D/V indication. The authors should indicate the orientation for each panel or provide a general orientation in the figure if all panels are oriented the same.

We have now added D/V or A/P indication to all figures.

Figure 23: "right view, left side" is confusing.

We have changed this to “ Each panel shows a ventral (left panel) and a left-side view (right panel).”

Line 406 : the first mention of the Platynereis cryptic segment, as far as I know, is Saudemont et al, 2008.

Thank you for pointing this out. We added the citation.

Figure 45: descending and decussating, 2nd and 3rd line of the legend.

Corrected

The format of data source tables is not homogeneized with some files in Excel format and others in plain comma format.

We have homogeneized the file formats of the supplements and source data. We have .csv files or .rds (R data format) files for the more complex data, such as tibble graphs that cannot be represented in a simple .csv format.